# On the Importance of Gaussianizing Representations

**Daniel Eftekhari** [1 2]   **Vardan Papyan** [3 1 2]

## Abstract

The normal distribution plays a central role in information theory – it is at the same time the best-case signal and worst-case noise distribution, has the greatest representational capacity of any distribution, and offers an equivalence between uncorrelatedness and independence for joint distributions. Accounting for the mean and variance of activations throughout the layers of deep neural networks has had a significant effect on facilitating their effective training, but seldom has a prescription for precisely what distribution these activations should take, and how this might be achieved, been offered. Motivated by the information-theoretic properties of the normal distribution, we address this question and concurrently present normality normalization: a novel normalization layer which encourages normality in the feature representations of neural networks using the power transform and employs additive Gaussian noise during training. Our experiments comprehensively demonstrate the effectiveness of normality normalization, in regards to its generalization performance on an array of widely used model and dataset combinations, its strong performance across various common factors of variation such as model width, depth, and training minibatch size, its suitability for usage wherever existing normalization layers are conventionally used, and as a means to improving model robustness to random perturbations.

## 1. Introduction

The normal distribution is unique – information theory shows that among all distributions with the same mean and variance, a signal following this distribution encodes the maximal amount of information (Shannon, 1948). This can be viewed as a desirable property in learning systems such as neural networks, where the activations of successive layers equivocates to successive representations of the data.

Moreover, a signal following the normal distribution is maximally robust to random perturbations (Cover & Thomas, 2006), and thus presents a desirable property for the representations of learning systems; especially deep neural networks, which are susceptible to random (Ford et al., 2019) and adversarial (Szegedy et al., 2014) perturbations. Concomitantly, the normal distribution is information-theoretically the worst-case perturbative noise distribution (Cover & Thomas, 2006), which suggests models gaining robustness to Gaussian noise should be robust to any other form of random perturbations.

We show that encouraging deep learning models to encode their activations using the normal distribution in conjunction with applying additive Gaussian noise during training, helps improve generalization. We do so by means of a novel layer – normality normalization – so-named because it applies the power transform, a technique used to gaussianize data (Box & Cox, 1964; Yeo & Johnson, 2000), and because it can be viewed as an augmentation of existing normalization techniques such as batch (Ioffe & Szegedy, 2015), layer (Ba et al., 2016), instance (Ulyanov et al., 2016), and group (Wu & He, 2018) normalization.

Our experiments comprehensively demonstrate the general effectiveness of normality normalization, in terms of its generalization performance, its strong performance across various common factors of variation such as model width, depth, and training minibatch size, which furthermore serve to highlight why it is effective, its suitability for usage wherever existing normalization layers are conventionally used, and its effect on improving model robustness under random perturbations.

In Section 2 we outline some of the desirable properties normality can imbue in learning models, which serve as motivating factors for the development of normality normalization. In Section 3 we provide a brief background on the power transform, before presenting normality normalization in Section 4. In Section 5 we describe our experiments, analyze the results, and explore some of the properties of models trained with normality normalization. In Section

[1]Department of Computer Science, University of Toronto, Toronto, Canada [2]Vector Institute, Toronto, Canada [3]Department of Mathematics, University of Toronto, Toronto, Canada. Correspondence to: Daniel Eftekhari <defte@cs.toronto.edu>.

*Proceedings of the 42$^{nd}$ International Conference on Machine Learning*, Vancouver, Canada. PMLR 267, 2025. Copyright 2025 by the author(s).

6 we comment on related work and discuss some possible future directions. Finally in Section 7 we contextualize normality normalization in the broader deep learning literature, and provide a few concluding remarks.

## 2. Motivation

In this section we present motivating factors for encouraging normality in feature representations in conjunction with using additive random noise during learning. Section 5 substantiates the applicability of the motivation through the experimental results.

### 2.1. Mutual Information Game & Noise Robustness

#### 2.1.1. Overview of the Framework

Under first and second moment constraints, the normal distribution is at the same time the best possible signal distribution, and the worst possible noise distribution; a result which can be studied in the context of the Gaussian channel (Shannon, 1948), and through the lens of the mutual information game (Cover & Thomas, 2006). In this framework, $X$ and $Z$ denote two independent random variables, representing the input signal and noise, and $Y = X + Z$ is the output. The mutual information between $X$ and $Y$ is denoted by $I(X;Y)$; $X$ tries to maximize this term, while $Z$ tries to minimize it. Both $X$ and $Z$ can encode their signal using any probability distribution, so that their respective objectives are optimized for.

Information theory answers the question of what distribution $X$ should choose to maximize $I(X;Y)$. It also answers the question of what distribution $Z$ should choose to minimize $I(X;Y)$. As shown by the following theorem, remarkably the answer to both questions is the same – the normal distribution.

**Theorem 2.1.** *(Cover & Thomas, 2006) Mutual Information Game. Let $X$, $Z$ be independent, continuous random variables with non-zero support over the entire real line, and satisfying the moment conditions $E[X] = \mu_x$, $E[X^2] = \mu_x^2 + \sigma_x^2$ and $E[Z] = \mu_z$, $E[Z^2] = \mu_z^2 + \sigma_z^2$. Further let $X^*$, $Z^*$ be normally distributed random variables satisfying the same moment conditions, respectively. Then the following series of inequalities holds*

$$
\begin{aligned}
I(X\ ;X\ + Z^*) &\le \\
I(X^*;X^* + Z^*) &\le \\
I(X^*;X^* + Z\ )&.
\end{aligned}
\tag{1}
$$

*Proof.* Without loss of generality let $\mu_x = 0$ and $\mu_z = 0$. The first inequality hinges on the entropy power inequality. The second inequality hinges on the maximum entropy of the normal distribution given first and second moment constraints. See Cover & Thomas (2006) for details.  □

This leads to the following minimax formulation of the game

$$
\min_Z \max_X I(X; X + Z) = \max_X \min_Z I(X; X + Z),
\tag{2}
$$

which implies that any deviation from normality, for $X$ or $Z$, is suboptimal from that player's perspective.

#### 2.1.2. Relation to Learning

How might this framework relate to the learning setting? First, previous works have shown that adding noise to the inputs (Bishop, 1995) or to the intermediate activations (Srivastava et al., 2014) of neural networks can be an effective form of regularization, leading to better generalization. Moreover, the mutual information game shows that, among encoding distributions with first and second moment constraints,[1] the normal distribution is maximally robust to random perturbations. Taken together these suggest that encoding activations using the normal distribution is the most effective way of using noise as a regularizer, because a greater degree of regularizing noise in the activations can be tolerated for the same level of corruption.

Second, the mutual information game suggests gaining robustness to Gaussian noise is optimal because it is the worst-case noise distribution. This suggests adding Gaussian noise – specifically – to activations during training should have the strongest regularizing effect. Moreover, gaining robustness to noise has previously been demonstrated to imply better generalization (Arora et al., 2018).

Finally, there exists a close correspondence between the mutual information between the input and the output of a channel subject to additive Gaussian noise, and the minimum mean-squared error (MMSE) in estimating the input given the output (Guo et al., 2005). This suggests that when Gaussian noise is added to a given layer's activations, quantifying the attenuation of the noise across the subsequent layers of the network, as measured by the mean-squared error (MSE) relative to the unperturbed activations, provides a measurable proxy for the mutual information between the activations of successive layers in the presence of noise.

### 2.2. Maximal Representation Capacity and Maximally Compact Representations

The entropy of a random variable is a measure of the number of bits it can encode (Shannon, 1948), and therefore of its representational capacity (Cover & Thomas, 2006). The normal distribution is the maximum entropy distribution for specified mean and variance. This suggests that a unit which encodes features using the normal distribution has maximal representation capacity given a fixed variance budget, and

---

[1]Note that conventional normalization layers respect precisely these constraints, since the mean and variance are used to normalize the pre-activations.

therefore encodes information as compactly as possible. This may then suggest that it is efficient for a unit (and by extension layer) to encode its activations using the normal distribution.

### 2.3. Maximally Independent Representations

Previous work has explored the beneficial effects of decorrelating features in neural networks (Huang et al., 2018; 2019; Pan et al., 2019). Furthermore, other works have shown that preventing feature co-adaptation is beneficial for training deep neural networks (Hinton et al., 2012).

For any set of random variables, for example representing the pre-activation values of various units in a neural network layer, uncorrelatedness does not imply independence in general. But for random variables whose marginals are normally distributed, then as shown by Lemma B.1, uncorrelatedness does imply independence when they are furthermore jointly normally distributed. Furthermore, for any given (in general, non-zero) degree of correlation between the random variables, they are maximally independent – relative to any other possible joint distribution – when they are jointly normally distributed.

We use these results to motivate the following argument: for a given level of correlation, encouraging normality in the feature representations of units would lead to the desirable property of maximal independence between them; in the setting where increased unit-wise normality also lends itself to increased joint normality.

## 3. Background: Power Transform

Before introducing normality normalization, we briefly outline the power transform (Yeo & Johnson, 2000) which our proposed normalization layer employs. Appendix C provides the complete derivation of the negative log-likelihood (NLL) objective function presented below.

Consider a random variable $H$ from which a sample $\boldsymbol{h} = \{h_i\}_{i=1}^N$ is obtained.[2] The power transform gaussianizes $\boldsymbol{h}$ by applying the following function for each $h_i$:

$$\psi(h; \lambda) = \begin{cases} \frac{1}{\lambda}\left((1+h)^\lambda - 1\right), & h \geq 0, \lambda \neq 0 \\ \log(1+h), & h \geq 0, \lambda = 0 \\ \frac{-1}{2-\lambda}\left((1-h)^{2-\lambda} - 1\right), & h < 0, \lambda \neq 2 \\ -\log(1-h), & h < 0, \lambda = 2 \end{cases}.$$
$$(3)$$

The parameter $\lambda$ is obtained using maximum likelihood

---

[2]In the context of normalization layers, $N$ represents the number of samples being normalized; for example in batch normalization, $N = BHW$ for convolutional layers, where $B$ is the minibatch size, and $H, W$ are respectively the height and width of the activation.

estimation, so that the transformed variable is as normally distributed as possible, by minimizing the following NLL:[3]

$$\mathcal{L}(\boldsymbol{h}; \lambda) = \frac{1}{2}\left(\log(2\pi) + 1\right) + \frac{1}{2}\log\left(\hat{\sigma}^2(\lambda)\right) \\ - \frac{\lambda - 1}{N}\sum_{i=1}^N \log(1 + h_i),$$
$$(4)$$

where $\hat{\mu}(\lambda) = \frac{1}{N}\sum_{i=1}^N \psi(h_i; \lambda)$ and $\hat{\sigma}^2(\lambda) = \frac{1}{N}\sum_{i=1}^N \left(\psi(h_i; \lambda) - \hat{\mu}(\lambda)\right)^2$.

## 4. Normality Normalization

To gaussianize a unit's pre-activations $\boldsymbol{h}$, normality normalization estimates $\hat{\lambda}$ using the method we present in Subsection 4.1, and then applies the power transform given by Equation 3. It subsequently adds Gaussian noise with scaling as described in Subsection 4.2. These steps are done between the normalization and affine transformation steps conventionally performed in other normalization layers.

### 4.1. Estimate of $\hat{\lambda}$

Differentiating Equation 4 w.r.t. $\lambda$ and setting the resulting expression to $0$ does not lead to a closed-form solution for $\hat{\lambda}$, which suggests an iterative method for its estimation; for example gradient descent, or a root-finding algorithm (Brent, 1971). However, motivated by the NLL's convexity in $\lambda$ (Yeo & Johnson, 2000), we use a quadratic series expansion for its approximation, which we outline in Appendix D.

With the quadratic form of the NLL, we can estimate $\hat{\lambda}$ with one step of the Newton-Raphson method:

$$\hat{\lambda} = 1 - \frac{\mathcal{L}'(\boldsymbol{h}; \lambda = 1)}{\mathcal{L}''(\boldsymbol{h}; \lambda = 1)},$$
$$(5)$$

where the series expansion has been taken around[4] $\lambda_0 = 1$. The expressions for $\mathcal{L}'(\boldsymbol{h}; \lambda = 1)$ and $\mathcal{L}''(\boldsymbol{h}; \lambda = 1)$ are outlined in Appendix D.

Appendix E provides empirical evidence substantiating the similarity between the NLL and its second-order series expansion around $\lambda_0 = 1$, and furthermore demonstrates the accuracy of obtaining the estimates $\hat{\lambda}$ using one step of the Newton-Raphson method.

Subsequent to estimating $\hat{\lambda}$, the power transform is applied to each of the pre-activations to obtain $x_i = \psi\left(h_i; \hat{\lambda}\right)$.

---

[3]To simplify the presentation, we momentarily defer the cases $\lambda = 0$ and $\lambda = 2$, and outline the NLL for $h \geq 0$ only, as the case for $h < 0$ follows closely by symmetry.

[4]The previously deferred cases of $\lambda = 0$ and $\lambda = 2$ are thus inconsequential, in the context of computing an estimate $\hat{\lambda}$, by continuity of the quadratic form of the series expansion for the NLL. However, these two cases still need to be considered when applying the transformation function itself.

We next discuss a few facets of the method.

**Justification for the Second Order Method**   The justification for using the Newton-Raphson method for computing $\hat{\lambda}$ is as follows:

- A first-order gradient-based method would require iterative refinements to its estimates of $\hat{\lambda}$ in order to find the minima, which would significantly affect runtime. In contrast, the Newton-Raphson method is guaranteed to find the minima of the quadratic loss in one step.

- A first-order gradient-based method for computing $\hat{\lambda}$ would require an additional hyperparameter for the step size. Due to the quadratic nature of the loss, the Newton-Raphson method necessarily does not require any such additional hyperparameter.

- The minibatch statistics $\hat{\mu}$ and $\hat{\sigma}^2$ are available in closed-form. It is therefore natural to seek a closed-form expression for $\hat{\lambda}$, which is facilitated by using the Newton-Raphson method.

**Location of Series Expansion**   The choice of taking the series expansion around $\lambda_0 = 1$ is justified using the following two complementary factors:

- $\hat{\lambda} = 1$ corresponds to the identity transformation, and hence having $\lambda_0 = 1$ as the point where the series expansion is taken, facilitates its recovery if this is optimal.

- It equivocates to assuming the least about the nature of the deviations from normality in the sample statistics, since it avoids biasing the form of the series expansion for the loss towards solutions favoring $\hat{\lambda} < 1$ or $\hat{\lambda} > 1$.

**Order of Normalization and Power Transform Steps**
Applying the power transform after the normalization step is beneficial, because having zero mean and unit variance activations simplifies several terms in the computation of $\hat{\lambda}$, as shown in Appendix D, and improves numerical stability.

**No Additional Learned Parameters**   Despite having increased normality in the features, this came at no additional cost in terms of the number of learnable parameters relative to existing normalization techniques.

**Test Time**   In the case where normality normalization is used to augment batch normalization, in addition to computing global estimates for $\mu$ and $\sigma^2$, we additionally compute a global estimate for $\lambda$. These are obtained using the respective training set running averages for these terms, analogously with batch normalization. At test time, these global estimates $\mu, \sigma^2, \lambda$ are used, rather than the test minibatch statistics themselves.

### 4.2. Additive Gaussian Noise with Scaling

Normality normalization applies regularizing additive random noise to the output of the power transform; a step which is also motivated through the information-theoretic principles described in Subsection 2.1, and whose regularizing effect is magnified by having gaussianized pre-activations.

For each input indexed by $i \in \{1, \ldots, N\}$, during training[5] we have $y_i = x_i + z_i \cdot \xi \cdot s$, where $x_i$ is the $i$-th input's post-power transform value, $z_i \sim \mathcal{N}(0, 1)$, $\xi \geq 0$ is the noise factor, and $s = \frac{1}{N} \|\boldsymbol{x} - \bar{\boldsymbol{x}}\|_1$ represents the zero-centered norm of the post-power transform values, normalized by the sample size $N$.

Importantly, scaling each of the sampled noise values $z_i$ for a given channel's minibatch[6] by the channel-specific scaling factor $s$, leads to an appropriate degree of additive noise for each of the channel's constituent terms $x_i$. This is significant because for a given minibatch, each channel's norm will differ from the norms of other channels.

Furthermore, we treat $s$ as a constant, so that its constituent terms are not incorporated during backpropagation.[7] This is significant because the purpose of $s$ is to scale the additive random noise by the minibatch's statistics, and not for it to contribute to learning directly by affecting the gradients of the constituent terms.

Note that we employ the $\ell_1$-norm for $\boldsymbol{x}$ rather than the $\ell_2$-norm because it lends itself to a more robust measure of dispersion (Pham-Gia & Hung, 2001).

Algorithm 1 provides a summary of normality normalization.

## 5. Experimental Results & Analysis

### 5.1. Experimental Setup

For each model and dataset combination, $M = 6$ models were trained, each with differing random initializations for the model parameters. Wherever a result is reported numerically, it is obtained using the mean performance and one standard error from the mean across the $M$ runs. The best performing models for a given dataset and model combination are shown in bold. Wherever a result is shown graphically, it is displayed using the mean performance, and its 95% confidence interval when applicable. The training

---

[5]We do not apply additive random noise with scaling at test time.

[6]For clarity the present discussion assumes the case where normality normalization is used to augment batch normalization. However, the discussion applies equally to other normalization layers, such as layer, instance, and group normalization.

[7]Implementationally, this is done by disabling gradient tracking when computing these terms.

**Algorithm 1** Normality Normalization

**Input:** $\boldsymbol{u} = \{u_i\}_{i=1}^N$

**Output:** $\boldsymbol{v} = \{v_i\}_{i=1}^N$

**Learnable Parameters:** $\gamma, \beta$

**Noise Factor:** $\xi \geq 0$

**Normalization**:
$$\hat{\mu} = \frac{1}{N} \sum_{i=1}^N u_i$$
$$\hat{\sigma}^2 = \frac{1}{N} \sum_{i=1}^N (u_i - \hat{\mu})^2$$
$$h_i = \frac{u_i - \hat{\mu}}{\sqrt{\hat{\sigma}^2 + \epsilon}}$$

**Power Transform and Scaled Additive Noise**:
$$\hat{\lambda} = 1 - \frac{\mathcal{L}'(\boldsymbol{h}; \lambda=1)}{\mathcal{L}''(\boldsymbol{h}; \lambda=1)}$$
$$x_i = \psi\left(h_i; \hat{\lambda}\right)$$

with gradient tracking disabled:
$$\bar{x} = \frac{1}{N} \sum_{i=1}^N x_i$$
$$s = \frac{1}{N} \sum_{i=1}^N |x_i - \bar{x}|$$

sample $z_i \sim \mathcal{N}(0, 1)$
$$y_i = x_i + z_i \cdot \xi \cdot s$$

**Affine Transform**:
$$v_i = \gamma \cdot y_i + \beta$$

configurations of the models[8] are outlined in Appendix F.

### 5.2. Generalization Performance

We evaluate layer normality normalization (LayerNormalNorm) and layer normalization (LayerNorm) on a variety of models and datasets, as shown in Table 1. A similar evaluation is done for batch normality normalization (BatchNormalNorm) and batch normalization (BatchNorm), shown in Table 2.

**Normality Normalization is Performant** LayerNormalNorm generally outperforms LayerNorm across multiple architectures and datasets, with a similar trend holding between BatchNormalNorm and BatchNorm.

**Effective With and Without Data Augmentations** Normality normalization is effective for models trained with (Table 1) and without (Table 2) data augmentations. This is of value in application areas such as time series analysis and fine-grained medical image analysis, where it is often not clear what data augmentations are appropriate.

[8]Code is made available at https://github.com/DanielEftekhari/normality-normalization.

*Table 1.* Validation accuracy across several datasets for a vision transformer (ViT) architecture (see training details for model specification), when using LayerNormalNorm (LNN) vs. LayerNorm (LN). Data augmentations were employed during training.

| DATASET | LN | LNN |
|---|---|---|
| SVHN | $94.61 \pm 0.31$ | $\mathbf{95.78 \pm 0.21}$ |
| CIFAR10 | $89.97 \pm 0.16$ | $\mathbf{91.18 \pm 0.13}$ |
| CIFAR100 | $66.40 \pm 0.42$ | $\mathbf{70.12 \pm 0.22}$ |
| FOOD101 | $73.25 \pm 0.19$ | $\mathbf{79.11 \pm 0.09}$ |
| IMAGENET TOP1 | $71.54 \pm 0.16$ | $\mathbf{75.25 \pm 0.07}$ |
| IMAGENET TOP5 | $89.40 \pm 0.11$ | $\mathbf{92.23 \pm 0.04}$ |

*Table 2.* Validation accuracy for several ResNet (RN) architecture and dataset combinations, when using BatchNormalNorm (BNN) vs. BatchNorm (BN). No data augmentations were employed during training.

| DATASET | MODEL | BN | BNN |
|---|---|---|---|
| CIFAR10 | RN18 | $88.89 \pm 0.07$ | $\mathbf{90.41 \pm 0.09}$ |
| CIFAR100 | RN18 | $62.02 \pm 0.17$ | $\mathbf{65.82 \pm 0.11}$ |
| STL10 | RN34 | $58.82 \pm 0.52$ | $\mathbf{63.86 \pm 0.45}$ |
| TINYIN TOP1 | RN34 | $58.22 \pm 0.12$ | $\mathbf{60.57 \pm 0.14}$ |
| TINYIN TOP5 | RN34 | $81.74 \pm 0.16$ | $\mathbf{83.31 \pm 0.13}$ |
| CALTECH101 | RN50 | $72.60 \pm 0.35$ | $\mathbf{74.71 \pm 0.51}$ |
| FOOD101 | RN50 | $61.15 \pm 0.44$ | $\mathbf{63.51 \pm 0.33}$ |

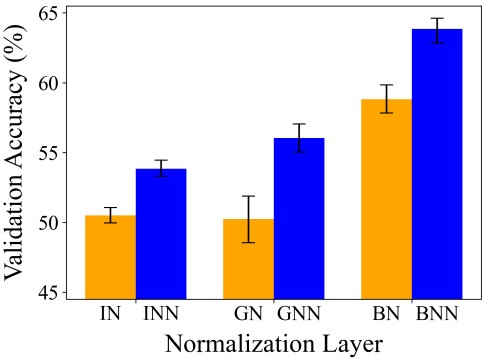

*Figure 1.* **Normality normalization is effective for various normalization layers.** Validation accuracy for ResNet34 architectures evaluated on the STL10 dataset. Each bar represents the performance of the ResNet34 architecture, when using the given normalization layer across the entire network. INN: InstanceNormalNorm, IN: InstanceNorm, GNN: GroupNormalNorm, GN: GroupNorm, BNN: BatchNormalNorm, BN: BatchNorm.

### 5.3. Effectiveness Across Normalization Layers

Figure 1 demonstrates the effectiveness of normality normalization across various normalization layer types. Here we further augmented group normalization (GroupNorm) to group normality normalization (GroupNormalNorm), and instance normalization (InstanceNorm) to instance normal-

ity normalization (InstanceNormalNorm).

Table 3 furthermore contrasts decorrelated batch normalization (Huang et al., 2018) with its augmented form decorrelated batch normality normalization, providing further evidence that normality normalization can be employed wherever normalization layers are conventionally used.

*Table 3.* As in Table 2, but for models using decorrelated Batch-NormalNorm (DBNN) vs. decorrelated BatchNorm (DBN).

| DATASET | MODEL | DBN | DBNN |
|---------|-------|-----|------|
| CIFAR10 | RN18 | $90.66 \pm 0.05$ | $\mathbf{91.50 \pm 0.03}$ |
| CIFAR100 | RN18 | $65.11 \pm 0.06$ | $\mathbf{67.53 \pm 0.10}$ |
| STL10 | RN34 | $66.76 \pm 0.29$ | $\mathbf{69.36 \pm 0.14}$ |

## 5.4. Effectiveness Across Model Configurations

**Network Width**  Figure 2 shows that BatchNormalNorm outperforms BatchNorm across varying WideResNet architecture model widths. Of particular note is that BatchNormalNorm shows strong performance even in the regime of relatively small network widths, whereas BatchNorm's performance deteriorates. This may indicate that for small-width networks, which do not exhibit the Gaussian process limiting approximation attributed to large-width networks (Neal, 1996; Lee et al., 2018; Jacot et al., 2018; Lee et al., 2019), normality normalization provides a correcting effect. This could, for example, be beneficial for hardware-limited deep learning applications.

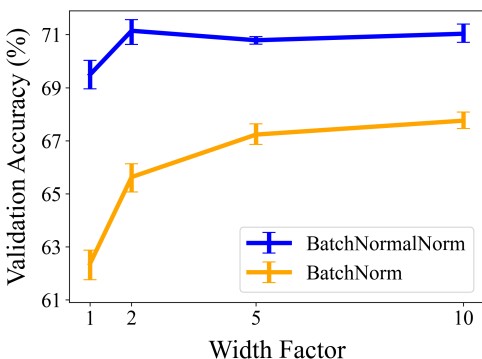

*Figure 2.* **Normality normalization is effective for small and large width networks.** Validation accuracy on the STL-10 dataset for WideResNet architectures with varying width factors when controlling for depth of 28, when using BatchNormalNorm vs. BatchNorm.

**Network Depth**  Figure 3 shows that BatchNormalNorm outperforms BatchNorm across varying model depths. This suggests normality normalization is beneficial both for small and large-depth models. Furthermore, the increased benefit to performance for BatchNormalNorm in deeper networks

suggests normality normalization may correct for an increased tendency towards non-normality as a function of model depth.

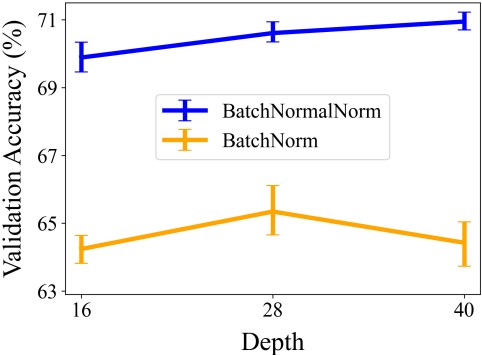

*Figure 3.* **Normality normalization is effective for networks of various depths.** Validation accuracy on the STL10 dataset for WideResNet architectures with varying depths when controlling for a width factor of 2, when using BatchNormalNorm vs. BatchNorm.

**Training Minibatch Size**  Figure 4 shows that BatchNormalNorm maintains a high level of performance across minibatch sizes used during training, which provides further evidence for normality normalization's general effectiveness across a variety of configurations.

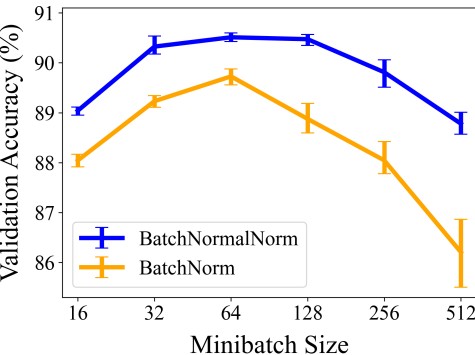

*Figure 4.* **Normality normalization is effective across minibatch sizes used during training.** Validation accuracy for ResNet18 architectures evaluated on the CIFAR10 dataset, with varying minibatch sizes used during training, when using BatchNormalNorm vs. BatchNorm.

## 5.5. Normality of Representations

Figure 5 shows representative Q–Q plots (Wilk & Gnanadesikan, 1968), a method for assessing normality, for post-power transform feature values when using BatchNormalNorm, and post-normalization feature values when using BatchNorm. Figure 6 shows an aggregate measure of normality across model layers, derived from several Q–Q plots corresponding to different channel and minibatch combinations. The figures correspond to models which have been

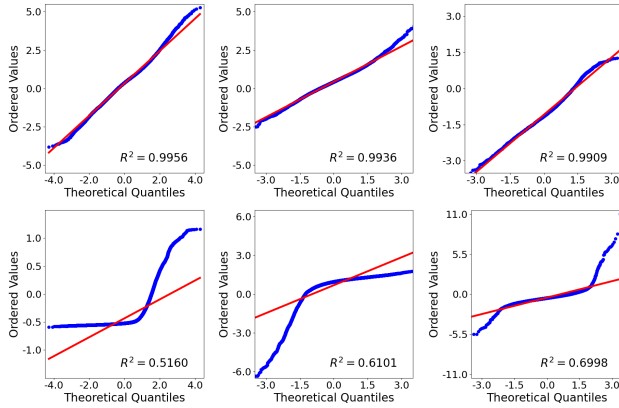

*Figure 5.* Representative Q–Q plots of feature values for models trained to convergence with BatchNormalNorm (post-power transform, top row) vs. BatchNorm (post-normalization, bottom row), measured for the same validation minibatch (ResNet34/STL10). Left to right: increasing layer number. The x-axis represents the theoretical quantiles of the normal distribution, and the y-axis the sample's ordered values. A higher $R^2$ value for the line of best fit signifies greater gaussianity in the features. BatchNormalNorm induces greater gaussianity in the features throughout the model, in comparison to BatchNorm.

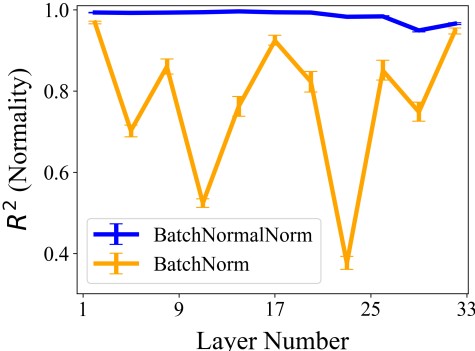

*Figure 6.* The average $R^2$ values for each model layer, derived from several Q–Q plots (see Figure 5) corresponding to 20 channel and 10 validation minibatch combinations, for models trained to convergence with BatchNormalNorm vs. BatchNorm. The plot demonstrates that normality normalization leads to greater gaussianity throughout the model layers.

trained to convergence. The plots demonstrate that normality normalization leads to greater gaussianity throughout the model layers.

### 5.6. Comparison of Additive Gaussian Noise With Scaling and Gaussian Dropout

Here we contrast the proposed method of additive Gaussian noise with scaling described in Subsection 4.2, with two other noise-based techniques.

The first is Gaussian dropout (Srivastava et al., 2014), where

for each input indexed by $i \in \{1, \ldots, N\}$, during training we have $y_i = x_i \cdot \left(1 + z_i \cdot \sqrt{\frac{1-p}{p}}\right)$, where $x_i$ is the $i$-th input's post-power transform value, $z_i \sim \mathcal{N}(0, 1)$, and $p \in (0, 1]$ is the retention rate.

The second is additive Gaussian noise, but without scaling by each channel's minibatch statistics. This corresponds to the proposed method in the case where $s$ is fixed to the mean of a standard half-normal distribution,[9] i.e. $s = \sqrt{\frac{2}{\pi}}$ across all channels; and thus does not depend on the channel statistics.

Figure 7 shows that additive Gaussian noise with scaling is more effective than Gaussian dropout, giving further evidence for the novelty and utility of the proposed method. It is also more effective than additive Gaussian noise (without scaling), which suggests the norm of the channel statistics plays an important role when using additive random noise.

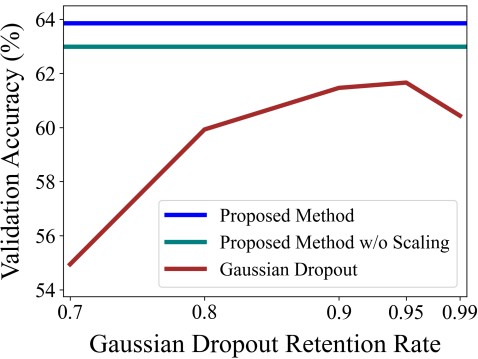

*Figure 7.* **Additive Gaussian noise with scaling is effective.** Validation accuracy for models trained with BatchNormalNorm (ResNet34/STL10), but with varying forms for the noise component of the normalization layer.

One reason why additive Gaussian noise with scaling may work better than Gaussian dropout, is because the latter scales activations multiplicatively, which means the effect of the noise is incorporated in the backpropagated errors. In contrast, the proposed method's noise component does not contribute to the gradient updates directly, because it is additive. This would suggest that models trained with normality normalization obtain higher generalization performance, because they must become robust to misattribution of gradient values during backpropagation, relative to the corrupted activation values during the forward pass.

### 5.7. Effect of Degree of Gaussianization

Here we consider what effect differing degrees of gaussianization have on model performance, as measured by the

---

[9]This value for $s$ precisely mirrors how it is calculated in Algorithm 1, since recall $s = \frac{1}{N} \sum_{i=1}^{N} |x_i - \bar{x}|$.

proximity of the estimate $\hat{\lambda}$ to the Newton-Raphson solution, which was given by Equation 5.

We control the proximity to the Newton-Raphson solution using a parameter $\alpha \in [0, 1]$ in the following equation

$$\hat{\lambda} = 1 - \alpha \frac{\mathcal{L}'(\boldsymbol{h}; \lambda = 1)}{\mathcal{L}''(\boldsymbol{h}; \lambda = 1)}, \qquad (6)$$

where $\alpha = 1$ corresponds to the Newton-Raphson solution, and decreasing values of $\alpha$ reduce the strength of the gaussianization.

Figure 8 demonstrates that the method's performance increases with increasing $\alpha$, and obtains its best performance for $\alpha = 1$. This provides further evidence that increasing gaussianity improves model performance.

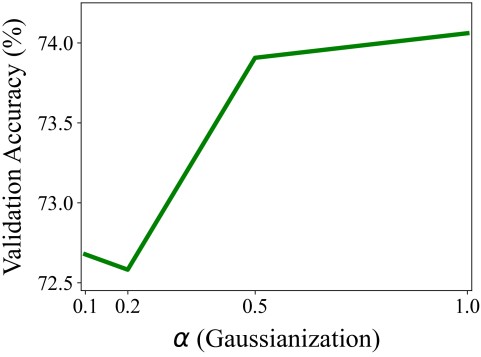

*Figure 8.* **Increasing gaussianity improves model performance.** Validation accuracy for models trained using BatchNormalNorm without noise (ResNet50/Caltech101), and with varying strengths for the gaussianization (parameterized by $\alpha$) when applying the power transform.

### 5.8. Controlling for the Power Transform and Additive Noise Components

Figure 9 demonstrates that both components of normality normalization – the power transform, and the additive Gaussian noise with scaling – each contribute meaningfully to the increase in performance for models trained with normality normalization.

### 5.9. Additional Experiments & Analysis

We next describe several additional experiments and analyses which serve to further demonstrate the effectiveness of normality normalization, and to substantiate the applicability of the motivation we presented in Section 2.

**Normality normalization induces robustness to noise at test time.** Appendix A.1 demonstrates that models trained using normality normalization are more robust to random noise at test time. This substantiates the applicability of the

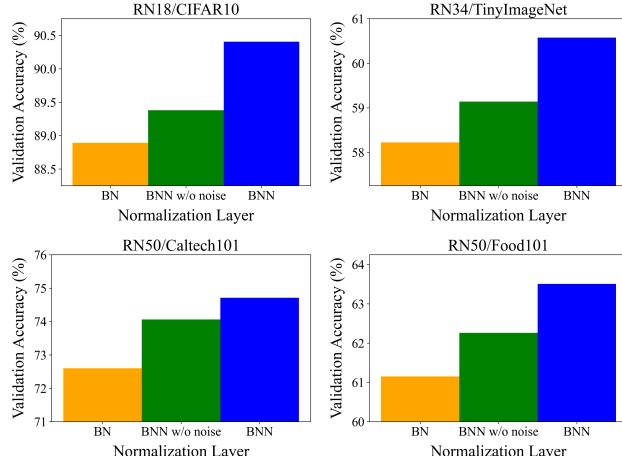

*Figure 9.* **Controlling for the effects of the power transform and the additive Gaussian noise with scaling components.** Each subplot shows the performance of models trained with BatchNormalNorm with the use of additive Gaussian noise with scaling (BNN), and without (BNN w/o noise), while using BatchNorm (BN) as a baseline. Subplot titles indicate the model and dataset combination.

noise robustness framework presented in Motivation Subsection 2.1, and consequently of the benefit of gaussianizing representations.

**Speed benchmarks.** Appendix A.2 shows that normality normalization increases runtime; with a larger deviation at training time than at test time.

**Normality normalization uniquely maintains gaussianity throughout training.** Appendix A.3 provides a graphical illustration of the fact that at initialization, layer preactivations are close to Gaussian regardless of the normalization layer employed; thus only models trained with normality normalization maintain gaussianity throughout training.

**Normality normalization induces greater feature independence.** Appendix A.4 demonstrates that normality normalization imbues models with greater joint normality and greater independence between channel features, throughout the layers of a model. This is of value in context of the benefit feature independence is thought to provide, which was explored in Motivation Subsection 2.3.

## 6. Related Work & Future Directions

**Power Transforms** Various power transforms have been developed (Box & Cox, 1964; Yeo & Johnson, 2000) and their properties studied (Hernandez & Johnson, 1980), for increasing normality in data. Box & Cox (1964) defined a power transform which is convex in its parameter, but is only defined for positive variables. Yeo & Johnson (2000)

presented an alternative power transform which was furthermore defined for the entire real line, preserved the convexity property with respect to its parameter for positive input values (concavity in the parameter for negative input values), and additionally addressed skewed input distributions.

It is worth noting that many power transforms were developed with the aim of improving the validity of statistical tests relying on the assumption of normality in the data. This is in contrast with the present work, which uses an information-theoretic motivation for gaussianizing.

**Gaussianization**    Alternative approaches to gaussianization, such as transformations for gaussianizing heavy-tailed distributions (Goerg, 2015), iterative gaussianization techniques (Chen & Gopinath, 2000; Laparra et al., 2011), and copula-based gaussianization (Nelsen, 2006), offer interesting directions for future work. Non-parametric techniques for gaussianizing, for example those using quantile functions (Gilchrist, 2000), may not be easily amenable to the deep learning setting where models are trained using backpropagation and gradient descent.

**Usage in Other Normalization Layers**    Works which have previously assumed normality in the pre-activations to motivate and develop their methodology, for example as seen in normalization propagation (Arpit et al., 2016), may benefit from normality normalization's explicit gaussianizing effect. It would also be interesting to explore what effect gaussianizing model weights might have, for example by using normality normalization to augment weight normalization (Salimans & Kingma, 2016).

**Adversarial Robustness**    It would be interesting to tie the present work with those suggesting robustness to $\ell_2$-norm constrained adversarial perturbations increases when training with Gaussian noise (Cohen et al., 2019; Salman et al., 2019). Furthermore, it has been suggested that adversarial examples and images corrupted with Gaussian noise may be related (Ford et al., 2019). This may indicate gaining robustness to Gaussian noise not only in the inputs, but throughout the model, can lead to greater adversarial robustness.

However, gaussianizing activations and training with Gaussian noise, may only be a defense in the distributional sense; exact knowledge of the weights (and consequently of the activation values), as is often assumed in the adversarial robustness setting, is not captured by the noise-based robustness framework, which is only concerned with distributional assumptions over the activation values. Nevertheless it does suggest that, on average, greater robustness may be attainable.

**Neural Networks as Gaussian Processes**    Neal (1996) showed that in the limit of infinite width, a single layer

neural network at initialization approximates a Gaussian process. This result has been extended to the multi-layer setting by Lee et al. (2018), and Jacot et al. (2018); Lee et al. (2019) suggest the Gaussian process approximation may remain valid beyond network initialization. However, these analyses still necessitate the infinite width limit assumption.

Subsequent work showed that batch normalization lends itself to a non-asymptotic approximation to normality throughout the layers of neural networks at initialization (Daneshmand et al., 2021). Given its gaussianizing effect, layers trained with normality normalization may be amenable – throughout training – to a non-asymptotic approximation to Gaussian processes. This could help further address the disparity in the analysis of neural networks in the infinite width limit, for example as in mean-field theory, with the finite width setting (Joudaki et al., 2023).

## 7. Conclusion

Among the methodological developments that have spurred the advent of deep learning, their success has often been attributed to their effect on the model's ability to learn and encode representations effectively, whether in the activations or in the weights. This can be seen, for example, by considering the importance attributed to initializing model weights suitably, or by the effect different activation functions have on learning dynamics.

Seldom has a prescription for precisely what distribution a deep learning model should use to effectively encode its activations, and exactly how this can be achieved, been investigated. The present work addresses this – first by motivating the normal distribution as the probability distribution of choice, and subsequently by materializing this choice through normality normalization.

It is perhaps nowhere clearer what representational benefit normality normalization provides, than when considering that no additional learnable parameters relative to existing normalization layers were introduced. This highlights – and precisely controls for the effect of – the importance of encouraging models to encode their representations effectively.

We presented normality normalization: a novel, principledly motivated, normalization layer. Our experiments and analysis comprehensively demonstrated the effectiveness of normality normalization, in regards to its generalization performance on an array of widely used model and dataset combinations, its consistently strong performance across various common factors of variation such as model width, depth, and training minibatch size, its suitability for usage wherever existing normalization layers are conventionally used, and through its effect on improving model robustness to random perturbations.

## Acknowledgments

We acknowledge the support provided by Compute Ontario (computeontario.ca) and the Digital Research Alliance of Canada (alliancecan.ca), and the support of the Natural Sciences and Engineering Research Council of Canada (NSERC), Discovery Grant RGPIN-2021-02527.

## Impact Statement

This work is of general interest to the machine learning and broader scientific community. There are many potential applications of the work, for which endeavoring to judiciously highlight one such possible application, in place of another, would be a precarious undertaking.

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

# A. Additional Experiments & Analysis

## A.1. Noise Robustness

We use the following framework to measure a model's robustness to noise (a similar setting is used by Arora et al. (2018)). For a given data point, consider a pair of units in a neural network, the first in the $k$-th layer and the second in the $\ell$-th layer. For the unit in the $k$-th layer, let $x$ denote the data point's post-normalization value. Let $\phi_{k,\ell}(x)$ be the same data point's post-normalization value for the unit in the subsequent layer $\ell$, where the function $\phi_{k,\ell}$ encapsulates all the intermediate computations between the two normalization layers $k$ and $\ell$.

Let $y = x + z \cdot \delta \cdot \frac{1}{N} \|\boldsymbol{x} - \bar{\boldsymbol{x}}\|_1$, where as in Subsection 4.2 $z \sim \mathcal{N}(0,1)$, $\delta \geq 0$, and here $\|\boldsymbol{x} - \bar{\boldsymbol{x}}\|_1$ represents a global estimate for the zero-centered norm of the post-normalized values, derived from the training set in its entirety. We then define noise robustness as follows:

*Table 4.* **Normality normalization induces robustness to noise at test time.** Evaluation of robustness to noise, using the relative error $\zeta_{k,\ell}^{\delta}$ for various layers $k$ and $\ell$, for various models trained with BatchNormalNorm and BatchNorm. Models were evaluated using noise factor $\delta = 0.5$. Top: ResNet18/CIFAR10, middle: ResNet18/CIFAR100, bottom: ResNet34/STL10. The layer $k$ at which noise is added is denoted on the left side of each row, and each column denotes a subsequent layer $\ell$. For each entry, $\zeta_{k,\ell}^{\delta}$ was averaged over the entire validation set, and over all channels in the $k$-th and $\ell$-th layers. This was subsequently averaged across $T = 6$ Monte Carlo draws for the random noise, and the values presented are furthermore the average across each of the $M = 6$ trained models. In each table entry, the top value represents the relative error for BatchNormalNorm (BNN), and the bottom value for BatchNorm (BN), with the best value shown in bold; lower is better. The tables provide evidence that models trained using normality normalization are generally more robust to random noise at test time.

| | | L5 | L9 | L13 | L17 |
|---|---|---|---|---|---|
| L1 | BNN | **0.044 ± 0.002** | **0.076 ± 0.002** | **0.124 ± 0.002** | **0.211 ± 0.005** |
| | BN | 0.160 ± 0.009 | 0.266 ± 0.014 | 0.390 ± 0.016 | 1.002 ± 0.032 |
| L5 | | | **0.031 ± 0.001** | **0.049 ± 0.001** | **0.080 ± 0.002** |
| | | | 0.060 ± 0.002 | 0.093 ± 0.004 | 0.201 ± 0.005 |
| L9 | | | | **0.048 ± 0.001** | **0.070 ± 0.001** |
| | | | | 0.060 ± 0.003 | 0.117 ± 0.003 |
| L13 | | | | | **0.142 ± 0.006** |
| | | | | | 0.203 ± 0.005 |

| | | L5 | L9 | L13 | L17 |
|---|---|---|---|---|---|
| L1 | BNN | **0.051 ± 0.001** | **0.076 ± 0.001** | **0.100 ± 0.001** | **0.387 ± 0.004** |
| | BN | 0.177 ± 0.007 | 0.333 ± 0.011 | 0.419 ± 0.021 | 1.922 ± 0.076 |
| L5 | | | **0.027 ± 0.002** | **0.038 ± 0.003** | **0.149 ± 0.012** |
| | | | 0.059 ± 0.009 | 0.073 ± 0.009 | 0.373 ± 0.041 |
| L9 | | | | **0.044 ± 0.001** | **0.151 ± 0.002** |
| | | | | 0.063 ± 0.003 | 0.249 ± 0.009 |
| L13 | | | | | **0.257 ± 0.001** |
| | | | | | 0.367 ± 0.020 |

| | | L9 | L17 | L25 | L33 |
|---|---|---|---|---|---|
| L1 | BNN | **0.373 ± 0.018** | **0.709 ± 0.034** | **0.452 ± 0.044** | **0.565 ± 0.053** |
| | BN | 0.615 ± 0.046 | 1.280 ± 0.080 | 1.900 ± 0.537 | 2.621 ± 0.257 |
| L9 | | | 0.141 ± 0.004 | **0.080 ± 0.003** | **0.099 ± 0.007** |
| | | | **0.120 ± 0.005** | 0.099 ± 0.011 | 0.307 ± 0.013 |
| L17 | | | | **0.102 ± 0.006** | **0.121 ± 0.011** |
| | | | | 0.104 ± 0.006 | 0.324 ± 0.012 |
| L25 | | | | | **0.051 ± 0.006** |
| | | | | | 0.120 ± 0.006 |

**Definition A.1** (Noise Robustness). For given realization of the noise sample $z$, let $\zeta_{k,\ell}^{\delta}(x,y)$-robustness be defined as

$$\zeta_{k,\ell}^{\delta}(x,y) := \frac{\|\phi_{k,\ell}(x) - \phi_{k,\ell}(y)\|_1}{\|\phi_{k,\ell}(x)\|_1}. \tag{7}$$

Thus $\zeta_{k,\ell}^{\delta}(x,y)$ measures the relative discrepancy between $\phi_{k,\ell}(x)$ and $\phi_{k,\ell}(y)$ when noise factor $\delta$ is used, and effectively represents the noise's attenuation from layer $k$ to layer $\ell$. Averaging $\zeta_{k,\ell}^{\delta}(x,y)$ over all data points, and over all units in the $k$-th and $\ell$-th layers, leads to a consolidated estimate of the noise robustness.

Table 4 demonstrates the increased robustness to noise obtained when using BatchNormalNorm in comparison to BatchNorm. This substantiates the applicability of the noise robustness framework presented in Motivation Subsection 2.1, and consequently of the benefit of gaussianizing representations in normality normalization.

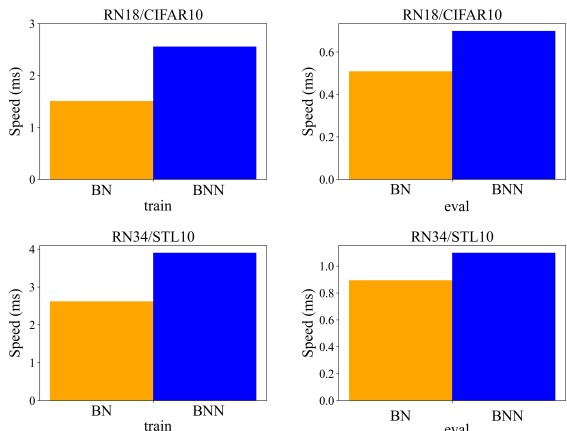

### A.2. Speed Benchmarks

Figure 10 shows the average per-sample running time for models using BatchNormalNorm and BatchNorm. The values are calculated by taking the average minibatch runtime at train/evaluation time, for the entire training/validation set, then normalizing by the number of samples in the minibatch. Values are obtained using an NVIDIA V100 GPU. For the purposes of these benchmarks, synchronization between the CPU and GPU was enforced.

The plots show that normality normalization increases runtime; with a larger deviation at training time than at test time. It is worth noting however, that the present work serves as a foundation, both conceptual and methodological, for future works which may continue to leverage the benefits of gaus-

*Figure 10.* Runtime comparison between models using BatchNormalNorm (BNN) and BatchNorm (BN) for two sets of model & dataset combinations; top: ResNet18/CIFAR10, bottom: ResNet34/STL10. The left hand plot shows the running time during training, and the right hand plot shows the running time during evaluation. See text for details.

sianizing. We believe improvements to the runtime of normality normalization can be obtained by leveraging approximations to the operations performed in the present form of normality normalization; for example the operations $\log(1 + h)$, and raising to the power.

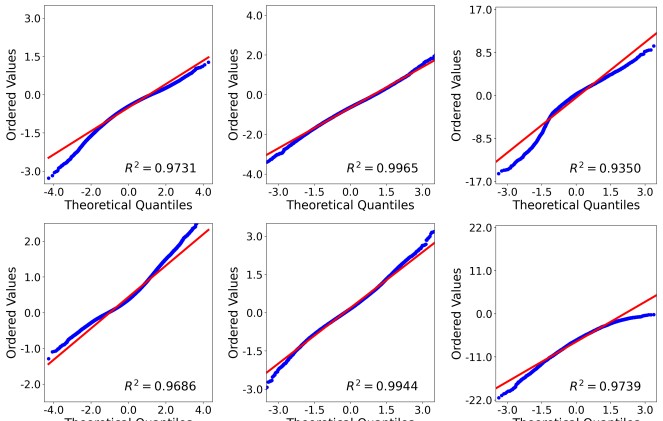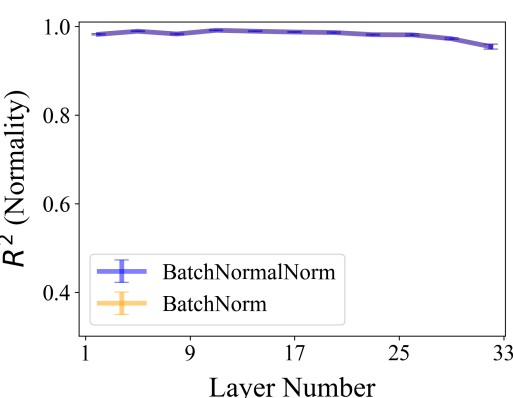

*Figure 11.* As in Figure 5 and Figure 6, but for networks at initialization. The plots provide a graphical illustration of the fact that at initialization, networks using either BatchNormalNorm or BatchNorm have close-to Gaussian pre-activations. However, as the networks are trained, BatchNormalNorm enforces and maintains normality while BatchNorm does not, as evidenced by Figure 6.

### A.3. Normality at Initialization

Figure 11 shows representative Q–Q plots, together with an aggregate measure of normality across model layers, for post-power transform feature values when using BatchNormalNorm, and post-normalization values when using BatchNorm, for models at initialization. It provides a graphical illustration of the fact that at initialization, the pre-activations are close to Gaussian regardless of the normalization layer employed; and thus that only the model trained with BatchNormalNorm enforces and maintains normality throughout training, as evidenced by Figure 6. Note that the Q–Q plots presented in Figure 5 and Figure 11 are obtained for the same corresponding minibatch and channel combinations.

### A.4. Joint Normality and Independence Between Features

Following the motivation we presented in Subsection 2.3, here we explore the potential effect normality normalization may have on increasing joint normality in the features, and the extent to which it may increase the independence between features.

We use the following experimental setup. For each layer of a ResNet34/STL10 model trained to convergence using either BatchNormalNorm or BatchNorm, we compute the correlation, joint normality, and mutual information over 10 pairs of channels, and across 10 validation minibatches.

We evaluate joint normality using the negative of the HZ-statistic (Henze & Zirkler, 1990) (higher values indicate greater joint normality), and evaluate independence using the adjusted mutual information (AMI) metric[10] (Vinh et al., 2010) (lower values indicate a greater degree of independence).

We evaluate joint normality across pairs of channels rather than across all of the channels in a layer, because measures of joint normality are sensitive to small deviations in sample statistics for finite sample sizes (Zhou & Shao, 2014; Ebner & Henze, 2020). Wherever we measure AMI, we use the square root of the number of sampled features as the number of bins (a generally accepted rule of thumb) when discretizing the features, and we use uniform binning, which is appropriate for (close to) normally distributed data.

Figure 12 demonstrates that models trained with BatchNormalNorm have higher joint normality, and have greater independence, across the model's layers. This is of value in context of the benefit feature independence is thought to provide, which was explored in Motivation Subsection 2.3.

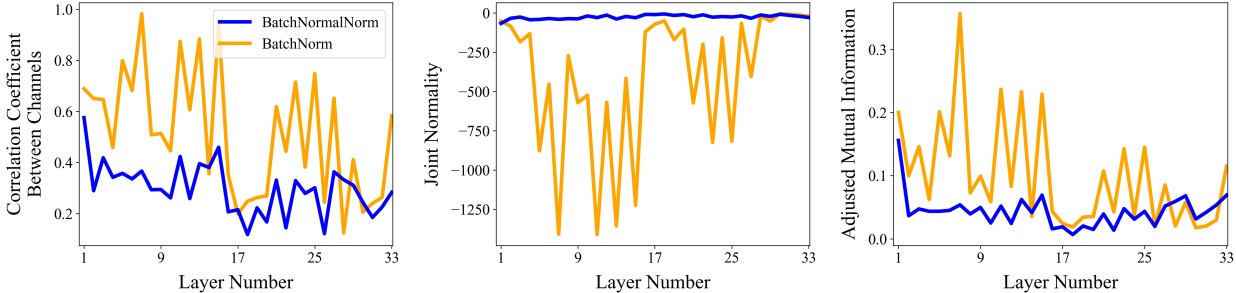

*Figure 12.* **Normality normalization induces greater feature independence.** Correlation, joint normality, and adjusted mutual information between pairs of channels for models trained to convergence using BatchNormalNorm vs. BatchNorm (ResNet34/STL10). The results are obtained by averaging the corresponding statistics across 10 channel pairs, and across 10 validation minibatches. Here joint normality is quantified using the negative of the HZ-statistic.

## B. Lemmas

**Lemma B.1.** *Bivariate Normality Minimizes Mutual Information. Let $X_1 \sim \mathcal{N}\left(x_1; \mu_1, \sigma_1^2\right)$ and $X_2 \sim \mathcal{N}\left(x_2; \mu_2, \sigma_2^2\right)$. Their mutual information $I\left(X_1; X_2\right)$ is minimized when the random variables are furthermore jointly normally distributed, i.e. $(X_1, X_2) \sim \mathcal{N}\left(\boldsymbol{x}; \boldsymbol{\mu}, \boldsymbol{\Sigma}\right)$, with $\boldsymbol{x} = \begin{bmatrix} x_1 \\ x_2 \end{bmatrix}$, $\boldsymbol{\mu} = \begin{bmatrix} \mu_1 \\ \mu_2 \end{bmatrix}$, $\boldsymbol{\Sigma} = \begin{bmatrix} \sigma_1^2 & \rho\sigma_1\sigma_2 \\ \rho\sigma_1\sigma_2 & \sigma_2^2 \end{bmatrix}$, and $\rho$ the correlation coefficient between $X_1, X_2$.*

---

[10]The AMI is a variation of mutual information, which adjusts for random chance. It is also bounded between 0 and 1, which makes it easier to interpret.

*Proof.* Consider two possible distributions, $f, g$, for the joint distribution over $(X_1, X_2)$, where $f$ denotes the probability density function (PDF) of the bivariate normal distribution, and $g$ can be any joint distribution. Our goal is to show that the mutual information between $X_1, X_2$, when they are distributed according to $g$, is lower-bounded by the mutual information between $X_1, X_2$ when they are distributed according to $f$.

For clarity of presentation, let the number of variables $f$ and $g$ take as arguments be clear from context, so that it is understood when they are used to denote their marginal distributions. Furthermore let $I_g(X_1; X_2)$ represent the mutual information when $(X_1, X_2)$ are distributed according to $g$, with the notation extending analogously to their joint $h_g(X_1, X_2)$ and marginal $h_g(X_1)$, $h_g(X_2)$ entropies under $g$.

We then have

$$
\begin{aligned}
I_g(X_1; X_2) &= h_g(X_1) + h_g(X_2) - h_g(X_1, X_2) \\
&= h_f(X_1) + h_f(X_2) - h_g(X_1, X_2) \\
&\geq h_f(X_1) + h_f(X_2) - h_f(X_1, X_2) \\
&= I_f(X_1; X_2) \\
&= \frac{1}{2}\log\left(2\pi e \sigma_1^2\right) + \frac{1}{2}\log\left(2\pi e \sigma_2^2\right) - \frac{1}{2}\log\left((2\pi e)^2\left(1 - \rho^2\right)\sigma_1^2\sigma_2^2\right) \\
&= \frac{1}{2}\log\left(\frac{1}{1 - \rho^2}\right),
\end{aligned}
\tag{8}
$$

where the second equality follows because by assumption the marginals are normally distributed, the inequality follows because the normal distribution maximizes entropy, and in the second-last equality we have used the expressions for the entropies of the univariate and bivariate normal distributions. $\qquad\square$

Consequently, when the random variables are jointly normally distributed, $\rho = 0$ implies $I(X_1; X_2) = 0$; thus uncorrelatedness implies independence.[11]

## C. Derivation of the Power Transform Negative Log-Likelihood

As in Section 3, consider a random variable $H$ from which a sample $\boldsymbol{h} = \{h_i\}_{i=1}^N$ is obtained. Recall that the power transform gaussianizes $\boldsymbol{h}$ by applying Equation 3 for each $h_i$, where the parameter $\lambda$ is obtained using maximum likelihood estimation, so that the transformed variable is as normally distributed as possible.

Denote the transformed random variable as $X \sim \mathcal{N}\left(x; \mu, \sigma^2\right)$, where $x = \psi(h; \lambda)$, and $\mu = \mu(\lambda)$, $\sigma^2 = \sigma^2(\lambda)$; i.e. $\mu$ and $\sigma^2$ are in general functions of $\lambda$. Obtaining the maximum likelihood estimates for $(\mu, \sigma^2, \lambda)$, shown next, requires evaluating the probability density function (PDF) of $H$, given the transformed variable $X$ is normally distributed.

Evaluating the cumulative distribution function (CDF) of $H$ gives[12]

$$
\begin{aligned}
F_H(h) &= P(H \leq h) \\
&= P\left((1 + \lambda X)^{1/\lambda} - 1 \leq h\right) \\
&= P\left(X \leq \frac{1}{\lambda}\left((1 + h)^{\lambda} - 1\right)\right) \\
&= F_X\left(\frac{1}{\lambda}\left((1 + h)^{\lambda} - 1\right)\right).
\end{aligned}
\tag{9}
$$

---

[11]The preceding result extends straightforwardly to the general multivariate setting, i.e. with more than two random variables.

[12]To simplify the presentation of the derivation, we omit the cases where $\lambda = 0$ and $\lambda = 2$, and outline the NLL for $h \geq 0$ only, as the case for $h < 0$ follows closely by symmetry.

Differentiating Equation 9 gives the PDF of $H$:

$$
\begin{aligned}
f_H\left(h; \mu\left(\lambda\right), \sigma^2\left(\lambda\right), \lambda\right) &= \frac{d}{dh} F_H\left(h\right) \\
&= \frac{d}{dh}\left(\frac{1}{2}\left(1 + \mathrm{erf}\left(\frac{\frac{1}{\lambda}\left(\left(1+h\right)^\lambda - 1\right) - \mu\left(\lambda\right)}{\sigma\left(\lambda\right)\sqrt{2}}\right)\right)\right) \\
&= \left(1+h\right)^{\lambda-1} \frac{1}{\sqrt{2\pi\sigma^2\left(\lambda\right)}} \exp\left(\frac{-1}{2\sigma^2\left(\lambda\right)}\left(\frac{\left(1+h\right)^\lambda - 1}{\lambda} - \mu\left(\lambda\right)\right)^2\right) \\
&= \left(1+h\right)^{\lambda-1} f_X\left(x; \mu\left(\lambda\right), \sigma^2\left(\lambda\right)\right).
\end{aligned}
\tag{10}
$$

The negative log-likelihood (NLL) of the sample according to this distribution is

$$
\begin{aligned}
\mathcal{L}&\left(\boldsymbol{h}; \mu\left(\lambda\right), \sigma^2\left(\lambda\right), \lambda\right) \\
&= -\frac{1}{N}\log\prod_{i=1}^{N} f_H\left(h_i; \mu\left(\lambda\right), \sigma^2\left(\lambda\right), \lambda\right) \\
&= -\frac{1}{N}\log\prod_{i=1}^{N}\left[\left(1+h_i\right)^{\lambda-1}\frac{1}{\sqrt{2\pi\sigma^2\left(\lambda\right)}}\exp\left(\frac{-1}{2\sigma^2\left(\lambda\right)}\left(x_i - \mu\left(\lambda\right)\right)^2\right)\right] \\
&= \frac{1}{2}\log\left(2\pi\right) + \frac{1}{2}\log\left(\sigma^2\left(\lambda\right)\right) + \frac{1}{2N\sigma^2\left(\lambda\right)}\sum_{i=1}^{N}\left(x_i - \mu\left(\lambda\right)\right)^2 - \frac{\lambda-1}{N}\sum_{i=1}^{N}\log\left(1+h_i\right).
\end{aligned}
\tag{11}
$$

Optimizing the NLL w.r.t. $\mu\left(\lambda\right)$ and $\sigma^2\left(\lambda\right)$ gives

$$
\begin{aligned}
\hat{\mu}\left(\lambda\right) &= \frac{1}{N}\sum_{i=1}^{N} x_i, \\
\hat{\sigma}^2\left(\lambda\right) &= \frac{1}{N}\sum_{i=1}^{N}\left(x_i - \hat{\mu}\left(\lambda\right)\right)^2.
\end{aligned}
\tag{12}
$$

Finally re-writing the NLL using the expressions for $\hat{\mu}\left(\lambda\right)$ and $\hat{\sigma}^2\left(\lambda\right)$, we obtain the profile NLL (Pickles, 1985):

$$
\mathcal{L}\left(\boldsymbol{h}; \hat{\mu}\left(\lambda\right), \hat{\sigma}^2\left(\lambda\right), \lambda\right) = \frac{1}{2}\left(\log\left(2\pi\right) + 1\right) + \frac{1}{2}\log\left(\hat{\sigma}^2\left(\lambda\right)\right) - \frac{\lambda-1}{N}\sum_{i=1}^{N}\log\left(1+h_i\right).
\tag{13}
$$

## D. Series Expansion of the Power Transform Loss

Let $\mathcal{L}_2\left(\boldsymbol{h}; \left(\lambda, \lambda_0 = 1\right)\right)$ denote the second-order series expansion of the power transform's negative log-likelihood (NLL) centered at $\lambda_0 = 1$, i.e.

$$
\mathcal{L}_2\left(\boldsymbol{h}; \left(\lambda, \lambda_0 = 1\right)\right) = \mathcal{L}\left(\boldsymbol{h}; \lambda = 1\right) + \left(\lambda - 1\right)\mathcal{L}'\left(\boldsymbol{h}; \lambda = 1\right) + \frac{\left(\lambda-1\right)^2}{2}\mathcal{L}''\left(\boldsymbol{h}; \lambda = 1\right).
\tag{14}
$$

We have[13]

$$
\mathcal{L}\left(\boldsymbol{h}; \lambda = 1\right) = \mathcal{L}\left(\boldsymbol{h}; \lambda\right)\Big|_{\lambda=1}
$$

$$
= \frac{1}{2}\log\left(2\pi + 1\right) + \frac{1}{2}\log\left(\hat{\sigma}^2\left(\lambda = 1\right)\right),
$$

$$
\mathcal{L}'\left(\boldsymbol{h}; \lambda = 1\right) = \frac{\partial \mathcal{L}\left(\boldsymbol{h}; \lambda\right)}{\partial \lambda}\Big|_{\lambda=1}
$$

$$
= \frac{1}{2\hat{\sigma}^2\left(\lambda = 1\right)}\frac{\partial \hat{\sigma}^2\left(\lambda\right)}{\partial \lambda}\Big|_{\lambda=1} - \frac{1}{N}\sum_{i=1}^{N}\log\left(1 + h_i\right),
$$

$$
\mathcal{L}''\left(\boldsymbol{h}; \lambda = 1\right) = \frac{\partial^2 \mathcal{L}\left(\boldsymbol{h}; \lambda\right)}{\partial \lambda^2}\Big|_{\lambda=1}
$$

$$
= \frac{-1}{2\left(\hat{\sigma}^2\left(\lambda = 1\right)\right)^2}\left(\frac{\partial \hat{\sigma}^2\left(\lambda\right)}{\partial \lambda}\Big|_{\lambda=1}\right)^2 + \frac{1}{2\hat{\sigma}^2\left(\lambda = 1\right)}\frac{\partial^2 \hat{\sigma}^2\left(\lambda\right)}{\partial \lambda^2}\Big|_{\lambda=1},
$$

(15)

where

$$
\frac{\partial \hat{\sigma}^2\left(\lambda\right)}{\partial \lambda} = \frac{2}{N}\sum_{i=1}^{N}\left[\left(\psi\left(h_i; \lambda\right) - \hat{\mu}\left(\lambda\right)\right)\left(\frac{\partial \psi\left(h_i; \lambda\right)}{\partial \lambda} - \frac{\partial \hat{\mu}\left(\lambda\right)}{\partial \lambda}\right)\right],
$$

(16)

$$
\therefore \frac{\partial \hat{\sigma}^2\left(\lambda\right)}{\partial \lambda}\Big|_{\lambda=1} = \frac{2}{N}\sum_{i=1}^{N}\left[\left(h_i - \hat{\mu}\left(\lambda = 1\right)\right)\left(\frac{\partial \psi\left(h_i; \lambda\right)}{\partial \lambda}\Big|_{\lambda=1} - \frac{\partial \hat{\mu}\left(\lambda\right)}{\partial \lambda}\Big|_{\lambda=1}\right)\right],
$$

(17)

with

$$
\frac{\partial \psi\left(h_i; \lambda\right)}{\partial \lambda}\Big|_{\lambda=1} = \left(1 + h_i\right)\left(\log\left(1 + h_i\right)\right) - h_i,
$$

$$
\frac{\partial \hat{\mu}\left(\lambda\right)}{\partial \lambda}\Big|_{\lambda=1} = \frac{1}{N}\sum_{i=1}^{N}\frac{\partial \psi\left(h_i; \lambda\right)}{\partial \lambda}\Big|_{\lambda=1},
$$

(18)

and

$$
\frac{\partial^2 \hat{\sigma}^2\left(\lambda\right)}{\partial \lambda^2} = \frac{2}{N}\sum_{i=1}^{N}\left[\left(\left(\psi\left(h_i; \lambda\right) - \hat{\mu}\left(\lambda\right)\right)\left(\frac{\partial^2 \psi\left(h_i; \lambda\right)}{\partial \lambda^2} - \frac{\partial^2 \hat{\mu}\left(\lambda\right)}{\partial \lambda^2}\right)\right)\right.
$$

$$
\left. + \left(\frac{\partial \psi\left(h_i; \lambda\right)}{\partial \lambda} - \frac{\partial \hat{\mu}\left(\lambda\right)}{\partial \lambda}\right)^2\right],
$$

(19)

$$
\therefore \frac{\partial^2 \hat{\sigma}^2\left(\lambda\right)}{\partial \lambda^2}\Big|_{\lambda=1} = \frac{2}{N}\sum_{i=1}^{N}\left[\left(\left(h_i - \hat{\mu}\left(\lambda = 1\right)\right)\left(\frac{\partial^2 \psi\left(h_i; \lambda\right)}{\partial \lambda^2}\Big|_{\lambda=1} - \frac{\partial^2 \hat{\mu}\left(\lambda\right)}{\partial \lambda^2}\Big|_{\lambda=1}\right)\right)\right.
$$

$$
\left. + \left(\frac{\partial \psi\left(h_i; \lambda\right)}{\partial \lambda}\Big|_{\lambda=1} - \frac{\partial \hat{\mu}\left(\lambda\right)}{\partial \lambda}\Big|_{\lambda=1}\right)^2\right],
$$

(20)

with

$$
\frac{\partial^2 \psi\left(h_i; \lambda\right)}{\partial \lambda^2}\Big|_{\lambda=1} = \left(1 + h_i\right)\left(\log\left(1 + h_i\right)\right)^2 - 2\frac{\partial \psi\left(h_i; \lambda\right)}{\partial \lambda}\Big|_{\lambda=1},
$$

$$
\frac{\partial^2 \hat{\mu}\left(\lambda\right)}{\partial \lambda^2}\Big|_{\lambda=1} = \frac{1}{N}\sum_{i=1}^{N}\frac{\partial^2 \psi\left(h_i; \lambda\right)}{\partial \lambda^2}\Big|_{\lambda=1}.
$$

(21)

Furthermore, because the power transform is applied after the normalization step (see main text), $\hat{\mu}\left(\lambda = 1\right) = 0$ and $\hat{\sigma}^2\left(\lambda = 1\right) = 1$.

## E. Evaluation of $\hat{\lambda}$ Estimates

Figure 13 provides representative examples substantiating the similarity between the negative log-likelihood (NLL) and its second-order series expansion around $\lambda_0 = 1$. The figure furthermore demonstrates the accuracy of obtaining estimates of $\hat{\lambda}$ using one step of the Newton-Raphson method.

---

[13]To simplify the presentation, we outline the series expansion for $h \geq 0$ only, as $h < 0$ follows closely by symmetry.

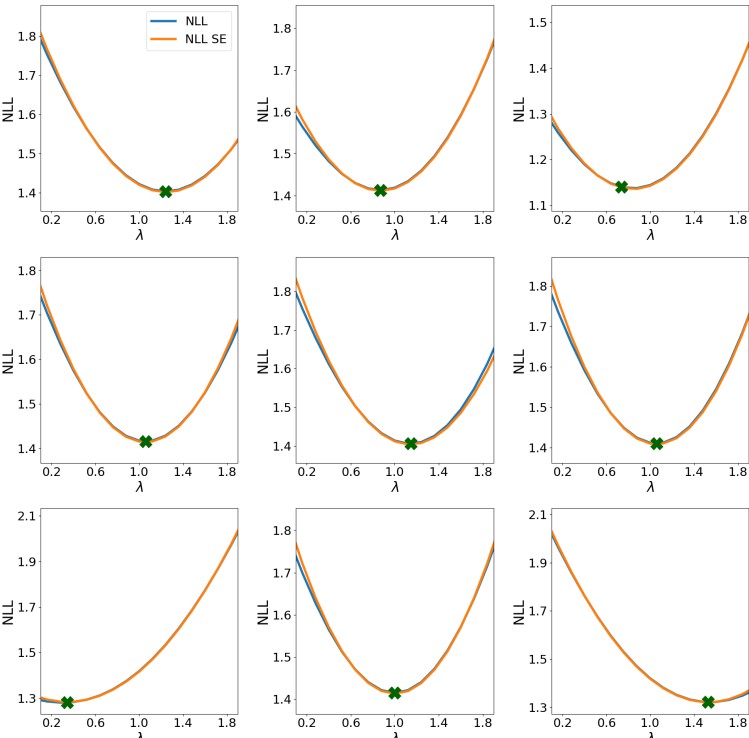

*Figure 13.* Normality normalization estimates for $\hat{\lambda}$ for a given training minibatch (ResNet18/CIFAR10). Left to right: increasing layer number. Top to bottom: estimates from various channels. Normality normalization's quadratic series expansion for the loss (NLL SE) closely approximates the original loss (NLL), leading to accurate estimates of $\hat{\lambda}$ (marked by ×).

## F. Training Details

### F.1. ResNet and WideResNet Experiments

The training configuration of the model and dataset combinations which use batch normality normalization (BatchNormalNorm/BNN), batch normalization (BatchNorm/BN), instance normality normalization (InstanceNormalNorm/INN), instance normalization (InstanceNorm/IN), group normality normalization (GroupNormalNorm/GNN), group normalization (GroupNorm/GN), decorrelated batch normality normalization (DBNN), and decorrelated batch normalization (DBN), are as follows.

We used a variety of residual network (ResNet) (He et al., 2016) and wide residual network (WideResNet) (Zagoruyko & Komodakis, 2016) architectures in our experiments. For all of the experiments except those using the TinyImageNet (TinyIN), Caltech101, and Food101 datasets, models were trained from random initialization for 200 epochs, with a factor of 10 reduction in learning rate at each 60-epoch interval. For the experiments using the TinyImageNet (TinyIN), Caltech101, and Food101 datasets, models were trained from random initialization for 100 epochs, with a factor of 10 reduction in learning rate at epochs $40, 70, 90$. A group size of 32 was used in all of the relevant group normalization experiments. For the Caltech101 dataset, each run used a random 90/10% allocation to obtain the training and validation splits respectively.[14] Each such run used its own unique random seed to generate the splits for that run, which facilitates greater precision in the reporting of our aggregate results across the runs.

In all of our experiments involving the ResNet18, ResNet34, and WideResNet architectures, stochastic gradient descent (SGD) with learning rate 0.1, weight decay $5 \times 10^{-4}$, momentum 0.9, and minibatch size 128 was used. In the experiments involving the ResNet50 architecture on the Caltech101 and Food101 datasets, SGD with learning rate 0.0125, weight decay $1 \times 10^{-4}$, momentum 0.9, and minibatch size 32 was used. A noise factor of $\xi = 0.4$, was used, as preliminary experiments

---

[14]The official Caltech101 dataset does not come with its own training/validation split.

demonstrated increases typically resulted in training instability. We also investigated several hyperparameter configurations, including for the learning rate, learning rate scheduler, weight decay, and minibatch size, across all the models and found the present configurations to generally work best across all of them.

### F.2. Vision Transformer Experiments

The training configuration of the model and dataset combinations which use layer normality normalization (LayerNormal-Norm/LNN) and layer normalization (LayerNorm/LN) are as follows. We used a vision transformer (Vaswani et al., 2017; Dosovitskiy et al., 2021) model consisting of 8 transformer layers, 8 attention heads, hidden dimension size of 768, and multi-layer perceptron (MLP) dimension size of 2304. A patch size of 4 was used throughout, except in the Food101 and ImageNet experiments, where it was set to 16.

For all of the experiments except those using the SVHN dataset, a learning rate warm-up strategy was employed, where the learning rate was linearly increased from a 0.1-th fraction of its value, to the full learning rate. After the warm-up phase, a cyclic learning rate schedule based on cosine annealing with periodic restarts was employed (Loshchilov & Hutter, 2017), with 50 iterations until the first restart, a factor of 2 for increasing the number of epochs between two warm restarts, and a minimal admissible learning rate of $1 \times 10^{-6}$. Models were trained using a minibatch size of 128, and for 900 epochs on the CIFAR10, CIFAR100, Food101 datasets, and for 200 epochs on the ImageNet dataset. For the ImageNet experiments, we applied weighted random sampling to sample training examples based on the training set's corresponding inverse class frequency for the data point; we found this to help across all the model configurations used. For the experiments involving the SVHN dataset, models were trained from random initialization for 200 epochs, with a factor of 10 reduction in learning rate at each 60-epoch interval, and a minibatch size of 32.

The AdamW optimizer (Kingma & Ba, 2015; Loshchilov & Hutter, 2019) with learning rate $1 \times 10^{-3}$, weight decay $5 \times 10^{-2}$, $(\beta_1, \beta_2) = (0.9, 0.999)$, $\epsilon = 1 \times 10^{-8}$ was used. A noise factor of $\xi = 1.0$ was used, as preliminary experiments demonstrated increases typically resulted in training instability. We also investigated several hyperparameter configurations, including for the learning rate, learning rate scheduler, weight decay, and minibatch size, across all the models and found the present configurations to generally work best across all of them.

The data augmentations used for the models presented in Table 1 are as follows. For the models trained on the SVHN dataset, mild random translations and rotations were used. For the models trained on the CIFAR10 and CIFAR100 datasets, random cropping, random horizontal flips, mild color jitters, and mixup (Zhang et al., 2018) were used. For the models trained on the Food101 dataset, random cropping with resizing, random horizontal flips, moderate color jitters, and mixup were used. For the models trained on the ImageNet dataset, random cropping with resizing, random horizontal flips, and moderate color jitters were used.

### F.3. Datasets and Frameworks

The datasets we used were CIFAR10, CIFAR100 (Krizhevsky, 2009), STL10 (Coates et al., 2011), SVHN (Netzer et al., 2011), Caltech101 (Li et al., 2022), TinyImageNet (Le & Yang, 2015), Food101 (Bossard et al., 2014), and ImageNet (Deng et al., 2009). We trained our models using the PyTorch (Paszke et al., 2019) machine learning framework.

