# OpenReview forum: "On the Importance of Gaussianizing Representations"
_ICML.cc/2025/Conference — ICML 2025 poster_

### Official Review · Reviewer_ps2s · 2025-02-18

**Overall Recommendation:** 4

**Summary:**

The authors propose adding a "Gaussianizing" step into normalisation layers such as batchnorm, which transforms the features so that they are approximately Gaussian-distributed. Specifically, they use the "power transform" originally proposed in the field of hypothesis testing, but propose approximating its objective by a quadratic so that the "correct" transform may be determined using a single Newton-Raphson step (which is important since this must be done every time a normalisation is used during a forward pass). The authors present many arguments, largely from an information-theoretic perspective, as to why having Gaussian features is desirable / important. This also motivates the authors' use of additive Gaussian noise as regularisation.

The authors present a wide range of experiments exploring the effect of their method on generalisation under different ablations, with models / datasets focusing on image classification tasks using ResNets and ViTs. They also verify the Gaussianity of the features when using their method vs standard normalisation layers. They find almost universal benefits to generalisation using their method, though, as detailed in a plot in the appendix, the proposed method does increase runtime by about 50% during training, and about 25% during testing.

### Update after rebuttals
I have maintained my score of "accept" - please see rebuttal comment

**Claims And Evidence:**

The main claim made by the paper is that forcing neural networks to have Gaussian distributed features throughout training can have desirable effects on performance, which is supported by their experiments. I have a couple of minor issues / questions about some of the experiments, which I will detail below.

**Essential References Not Discussed:**

I am not aware of any essential references that have not been discussed.

**Experimental Designs Or Analyses:**

The experiments only consider image classification tasks using ResNets and ViTs, though this includes a variety of architectures and datasets, and are run from multiple random seeds with error bars given. In an ideal world, the authors would have evaluated their method in another domain, such as language modelling with transformers, though given the extensive ablations and other investigations, I think this is an appropriate amount of experiments for a conference paper.

In the ResNet experiments, Table 2 says that data augmentation was not used. Data augmentation is usually used for ResNet experiments on datasets like CIFAR-10 and CIFAR-100, and greatly improves performance (e.g. we would expect closer to 93 or 94% for a ResNet18 on CIFAR-10 using data augmentation, rather than the 89% achieved by the paper's baseline model). This is especially strange given that Table 1 says that data augmentation **was** used in the ViT experiments. I would have preferred to see data augmentation for the ResNet experiments, as this is standard practice.

**Methods And Evaluation Criteria:**

Yes. The hypothesis is that Gaussianizing representations could have positive effects on model performance. The proposed method effectively Gaussianises the features (as shown in Figure 5) and results in an improvement to validation accuracy on realistic tasks (Figure 1, Table 1, Table 2, etc.).

**Other Comments Or Suggestions:**

- It would be useful if the authors state at the start of section 3.2 that the additive Gaussian noise is an additional regularisation technique enabled / enhanced by the Gaussianisation of the features, and not actually part of the Gaussianisation process itself (assuming I have understood this correctly). I was confused by this while reading the paper.
- Related to the previous point, I think it'd be nice to have figure 9 (comparing base model vs. base + gaussianised features vs base + gaussianised features + additive gaussian noise regularisation) in the main text. The Motivation section (section 5) could probably be shortened to accommodate this.
- Please state in the main text a briefly summarised version of the speed results, e.g. "the method increases training time by roughly 50%" etc. It'd also be nice to briefly explain what specific part of the algorithm causes such large slowdowns.
- The authors may be interested to try their method on some "speedrun" benchmarks e.g. https://github.com/KellerJordan/cifar10-airbench or https://github.com/KellerJordan/modded-nanogpt. Obviously the wallclock time will not be impressive given the method's current slowdowns, but loss vs iteration / epoch would be interesting.

**Other Strengths And Weaknesses:**

**Strengths**
- The paper is well written. In particular, the connections to information theory and motivations are clearly and extensively explained.
- I have not seen any work that explicitly Gaussianises the features in a neural network before, so as far as I'm aware, this is novel.
- The suggested method does offer convincing improvements in validation accuracy
- The authors have included a wide range of relevant ablations, analyses, and justifications, either in the main text or the appendix, which answered many questions I had while reading the paper.
- The authors identify that, when your features are Gaussian, decorrelating / whitening transformations actually imply independent features. This could have important implications for recent works that utilise whitening / decorrelating transformations in optimisation, e.g. https://kellerjordan.github.io/posts/muon/ or https://arxiv.org/abs/2412.13148.
- The authors try their method on 4 different types of normalisation (batchnorm, layernorm, instancenorm, and group norm) and show improvements in all cases.

**Weaknesses**
- The ResNet experiments do not use data augmentation, which is standard practice, and no explanation is given as to why.
- The specific implementation proposed in this paper increases runtime by  25-50%, which is quite significant.
- Some of the technical detail weren't very clearly explained. Specifically, slightly more explanation of where the NLL for the power transform comes from would make the paper easier to read. Also, it is not entirely clear what the QQ plots in the main paper (Fig 5 and 6) are showing (e.g. which layers are you plotting for, where does each value on the plot come from).

**Questions For Authors:**

- Why did you not use data augmentation on the ResNets? Can you rerun these experiments with data augmentation and include them in the paper?
- In Section 5.1.2. I didn't fully understand the point about (line 350) about not wanting to corrupt the distributions when regularising. Is corruption not the point of random noise / regularisation like dropout?

**Relation To Broader Scientific Literature:**

The paper proposes a modification that can be applied to any neural network which utilises normalisation layers, and therefore has very broad scope. Moreover, the specific algorithm proposed in this paper is less important than the proof of concept that Gaussianing features in normalisation layers is beneficial. Further refinements or alternative algorithms to efficiently Gaussianise features in neural networks could be developed in the future if required.

**Theoretical Claims:**

The paper does not make any theoretical claims (e.g. theorems, bounds). Derivations for the formulas used by their method are given in appendix C, which I did not check.

---

> ### Author Rebuttal · Authors · 2025-03-31
>
> Dear Reviewer ps2s,
>
> We address all of your comments below.
> >
> >Regarding the use of data augmentations in the ResNet experiments.
> >
> We have run an experiment to verify the performance of ResNet18 x CIFAR10 using BatchNormalNorm (BNN), and contrast this with a well-documented baseline - which is in line with your expected level of performance when using BatchNorm (BN) (https://github.com/kuangliu/pytorch-cifar). We use the same data augmentations of transforms.RandomCrop(32, padding=4) and transforms.RandomHorizontalFlip() as listed in the repository. Across $M=6$ runs, we obtained a mean validation set accuracy of $94.93$%$\pm0.05$, which surpasses the reference performance of $93.02$% listed in the repository; this latter figure being in line with your expected level of performance for BN when data augmentations are employed. This serves to demonstrate that BNN continues to scale and outperform with the use of data augmentations in the ResNet experiments; and is analogous to the findings for LayerNormalNorm (LNN) and LayerNorm (LN).
> >
> >Regarding the justification for not employing data augmentations in Table 2.
> >
> We next provide our justification for this choice in our experimental design. Our goal was for the set of experiments with data augmentations (Table 1) and without data augmentations (Table 2) to serve as an ablation - showing that the method performs well regardless of specific augmentation techniques used.
>
> Crucially: in many application areas, such as in time series analyses, and in fine-grained medical imaging tasks, it is often not clear what data augmentations are appropriate. Therefore, we believe demonstrating that our method performs strongly relative to other normalization layers - with and without the use of data augmentations - is extremely valuable.
> >
> >Regarding clarifying Figures 5 & 6.
> >
> Figure 5 shows the following: In a given layer of a neural network, we take all the post-normalization features for a given channel and minibatch combination. We then compute a QQ-plot and its associated $R^{2}$ value for the line of best fit. Now consider such a plot for three layers at various depths in the network. Thus Figure 5 serves to demonstrate graphically that normality normalization leads to higher normality in the features, as demonstrated by the higher $R^2$ values for the line of best fit. Crucially then, Figure 6 then substantiates these findings quantitatively: for each layer of the neural network (x-axis), we take $200$ QQ-plots corresponding to $20$ channels and $10$ validation minibatch combinations, we compute the $R^{2}$ values for each of these $200$ QQ-plots, then plot the mean $R^{2}$ value for that layer. Thus the figure demonstrates that throughout the layers of a network, normality normalization leads to much higher normality.
> >
> >"slightly more explanation of where the NLL for the power transform comes from"
> >
> Please see the paragraph about the NLL in our response to Reviewer Eseb, which precisely addresses your comment here.
> >
> >Regarding clarifying that Gaussian noise is an additional regularization technique.
> >
> We believe this is an excellent suggestion and have modified the paper accordingly.
> >
> >Regarding the inclusion of Figure 9 in the main paper.
> >
> We have made this change now - the camera-ready version of the paper affords an additional page (9 instead of 8) for the main text, thereby making its inclusion very natural.
> >
> >Regarding summarizing the speed results in the main text and describing what parts of the algorithm cause slowdowns.
> >
> We have now commented in Section 4.6 Additional Experiments & Analysis, under paragraph Speed Benchmarks, your suggested note on the speed difference.
>
> The main speed differences occur due to the operations log(1+x) ("log1p") and raising to the power. During the work, we investigated making series expansion approximations to these operations, and substantiating their efficacy is a promising direction for future work.
> >
> >"In Section 5.1.2. I didn't fully understand the point about (line 350) about not wanting to corrupt the distributions when regularising."
> >
> Here we are suggesting the following subtle distinction: we want to be able to add as much regularizing noise as possible, but without fundamentally corrupting the underlying signal. The key being that when using Gaussian encodings, the threshold for the amount of noise we can add is higher. You may also find the reference (Guo et al. 2005) (also referenced in our manuscript) to be of interest.
>
> We have furthermore added experimental results contrasting decorrelated BatchNorm (DBN) with decorrelated BatchNormalNorm (DBNN); please see the thread with Reviewer 7TPa. These experimental results provide further evidence for the strong performance of normality normalization across various normalization layers.
>
> We believe we have comprehensively addressed your comments here. We would be highly appreciative if you would consider increasing the score for our submission; thank you.

---

> > ### Comment · Reviewer_ps2s · 2025-04-02
> >
> > Thank you for your response. My concerns have been adequately addressed. I would suggest putting the explanation you have me of Figures 5 and 6 somewhere in the paper, if it is not already in there. Whilst other reviewers have raised some concerns and interesting points about related work and adversarial robustness, these are all minor in my opinion, so I would like to maintain my original score of "Accept", and emphasise that I think this is very interesting work that could spur further investigation into (efficiently) Gaussianising representations.

---

### Official Review · Reviewer_7TPa · 2025-03-12

**Overall Recommendation:** 4

**Summary:**

Full disclosure: I was a reviewer for a paper for ICLR 2025 which seems to be largely mirroring this paper and I assume that this is a resubmission of that paper (I'm reviewer G4ZL here: https://openreview.net/forum?id=9ut3QBscB0)


This paper introduces normality normalization as a  new type of normalization layer that attempts to impose stronger Gaussianity on the activations in neural networks. They motivate the via information theory by invoking classical facts about Gaussian distribution, namely that Gaussian is the “best-case signal” and “worst-case noise,” so adopting Gaussian representations plus Gaussian noise can maximize information capacity and tolerance to perturbations.  The key technical part is applying Yeo-Johnson power transform on the normalized activations and making them marginally closer to normal (marginal likelihood). They also propose an additive Gaussian noise.  Since this is only a way of correcting statistics, this can be applied to both Layer and Batch Norm to create “BatchNormalNorm,” “LayerNormalNorm,” etc.

In experiments on CIFAR-10/100, SVHN, STL10, TinyImageNet, Caltech101, Food101, and ImageNet, for ResNets, WideResNets, & Vision Transformers), they show normality normalization outperforms standard normalization (Tables 1 & 2). The authors also claim enhanced robustness to random noise via quantitative attenuation metrics.

## Update after rebuttal

After reviewing the rebuttal responses, am happy to increase my score from 3 to 4(Accept). I am happy with the main paper and the explanations provided here, and consider this paper to be a good contribution for ICML.

**Claims And Evidence:**

While the central claim that normality normalization outperforms the classical normalization seems to be validated by results in Table 1 & 2, there seems to be some gap between reported values for baselines and those reported in other papers. This leaves me the impression that the experimental setup (hyper params and training ) for the baselines may be not good enough to fully substantiate the empirical claims made in the paper. For example, I adapted the code found here: https://www.kaggle.com/code/kmldas/cifar10-resnet-90-accuracy-less-than-5-min, for a small resnet, and even the no data augmented reaches 90.25% accuracy. I understand that this is not the same ResNet they used, but the fact that a small ResNet can achieve >90% accuracy without data augmentation, makes the improvement to 90.4% in Table1 when using BNN, somewhat less significant. I'm eager to hear what authors have to clarify this.

the sample code I used :
```python
import os
import torch
import torch.nn as nn
import torch.nn.functional as F
import torchvision.transforms as tt
from torchvision.datasets import CIFAR10
from torch.utils.data import DataLoader
import time

# Device setup
device = torch.device('cuda' if torch.cuda.is_available() else 'cpu')
def to_device(x, device): return x.to(device, non_blocking=True) if isinstance(x, torch.Tensor) else [to_device(item, device) for item in x]

class DeviceDataLoader:
    def __init__(self, dl, device): self.dl, self.device = dl, device
    def __iter__(self): return (to_device(batch, self.device) for batch in self.dl)
    def __len__(self): return len(self.dl)

# Model definition
def accuracy(outputs, labels):
    _, preds = torch.max(outputs, dim=1)
    return torch.tensor(torch.sum(preds == labels).item() / len(preds))

def conv_block(in_ch, out_ch, pool=False):
    layers = [nn.Conv2d(in_ch, out_ch, kernel_size=3, padding=1), nn.BatchNorm2d(out_ch), nn.ReLU(inplace=True)]
    if pool: layers.append(nn.MaxPool2d(2))
    return nn.Sequential(*layers)

class ResNet9(nn.Module):
    def __init__(self, in_channels, num_classes):
        super().__init__()
        self.conv1 = conv_block(in_channels, 64)
        self.conv2 = conv_block(64, 128, pool=True)
        self.res1 = nn.Sequential(conv_block(128, 128), conv_block(128, 128))
        self.conv3 = conv_block(128, 256, pool=True)
        self.conv4 = conv_block(256, 512, pool=True)
        self.res2 = nn.Sequential(conv_block(512, 512), conv_block(512, 512))
        self.classifier = nn.Sequential(nn.MaxPool2d(4), nn.Flatten(), nn.Linear(512, num_classes))

    def forward(self, x):
        x = self.conv1(x)
        x = self.conv2(x)
        x = self.res1(x) + x
        x = self.conv3(x)
        x = self.conv4(x)
        x = self.res2(x) + x
        return self.classifier(x)

    def training_step(self, batch):
        images, labels = batch
        out = self(images)
        return F.cross_entropy(out, labels)

    def validation_step(self, batch):
        images, labels = batch
        out = self(images)
        loss = F.cross_entropy(out, labels)
        acc = accuracy(out, labels)
        return {'val_loss': loss.detach(), 'val_acc': acc}

    def validation_epoch_end(self, outputs):
        batch_losses = [x['val_loss'] for x in outputs]
        epoch_loss = torch.stack(batch_losses).mean()
        batch_accs = [x['val_acc'] for x in outputs]
        epoch_acc = torch.stack(batch_accs).mean()
        return {'val_loss': epoch_loss.item(), 'val_acc': epoch_acc.item()}

# Training functions
@torch.no_grad()
def evaluate(model, val_loader):
    model.eval()
    outputs = [model.validation_step(batch) for batch in val_loader]
    return model.validation_epoch_end(outputs)

def train_model(epochs, max_lr, model, train_dl, valid_dl, weight_decay=1e-4, grad_clip=0.1):
    optimizer = torch.optim.Adam(model.parameters(), max_lr, weight_decay=weight_decay)
    scheduler = torch.optim.lr_scheduler.OneCycleLR(optimizer, max_lr, epochs=epochs, steps_per_epoch=len(train_dl))

    history = []
    start_time = time.time()

    for epoch in range(epochs):
        # Training
        model.train()
        train_losses = []
        lrs = []

        for batch in train_dl:
            loss = model.training_step(batch)
            train_losses.append(loss)
            loss.backward()

            # Gradient clipping
            if grad_clip:
                nn.utils.clip_grad_value_(model.parameters(), grad_clip)

            optimizer.step()
            optimizer.zero_grad()

            # Record & update learning rate
            lrs.append(optimizer.param_groups[0]['lr'])
            scheduler.step()

        # Validation
        result = evaluate(model, valid_dl)
        result['train_loss'] = torch.stack(train_losses).mean().item()
        result['lrs'] = lrs

        # Print progress
        print(f"Epoch [{epoch}], lr: {lrs[-1]:.5f}, train_loss: {result['train_loss']:.4f}, val_loss: {result['val_loss']:.4f}, val_acc: {result['val_acc']:.4f}")
        history.append(result)

    train_time = time.time() - start_time
    print(f"Training completed in {train_time/60:.2f} minutes")
    return history

if __name__ == "__main__":
    stats = ((0.4914, 0.4822, 0.4465), (0.2023, 0.1994, 0.2010))
    train_tfms = tt.Compose([tt.ToTensor(), tt.Normalize(*stats)])
    valid_tfms = tt.Compose([tt.ToTensor(), tt.Normalize(*stats)])

    # Dataset
    os.makedirs(data_dir, exist_ok=True)
    train_ds = CIFAR10(root= './data', train=True, download=True, transform=train_tfms)
    valid_ds = CIFAR10(root= './data', train=False, download=True, transform=valid_tfms)

    # DataLoaders
    batch_size = 400
    train_dl = DeviceDataLoader(DataLoader(train_ds, batch_size, shuffle=True, num_workers=3, pin_memory=True), device)
    valid_dl = DeviceDataLoader(DataLoader(valid_ds, batch_size*2, num_workers=3, pin_memory=True), device)

    # Model
    model = ResNet9(3, 10).to(device)

    # Training
    history = train_model(
        epochs=15,
        max_lr=0.01,
        model=model,
        train_dl=train_dl,
        valid_dl=valid_dl,
        weight_decay=1e-4,
        grad_clip=0.1
    )

```

**Essential References Not Discussed:**

- Weight Normalization (Salimans & Kingma)
- Filter Response Normalization (Singh & Krishnan)
- Normalization Propagation (Arpit 2016)
- While not Copula-based or rank-based inverse normal transforms as direct ways to Gaussianize data, or Lambert W transformations for heavy-tailed distributions
- Copula-Based Gaussianization – In statistics, any multivariate data can be transformed to have normal marginals by using a copula. A Gaussian copul method assumes the data can be mapped into a joint Gaussian via monotonic marginal CDF mappings.
While the paper focuses on certain transforms, it doesn’t mention the general copula framework, which might be interesting

**Experimental Designs Or Analyses:**

The experimental setup ticks the basic
- Cross validation
- They use 6 runs with independent seeds to obtain mean and standard error
- They ablate key pieces: the power transform alone, noise alone, partial transform strength.
- They measure Q–Q plots to show actual Gaussianity, which is a key claim of the proposed layer
However, as mentioned in the Claims And Evidence section, I have some reservations about the optimality of the baseline results and would be eager to hear clarifications from authors

**Methods And Evaluation Criteria:**

Yes. They evaluate classification performance on mainstream benchmarks (CIFAR, SVHN, TinyImageNet, etc.), measure standard top-1 (and occasionally top-5) accuracy, and compare to widely used baselines (BatchNorm, LayerNorm, etc.). They also run controlled studies on batch size, network width, and depth, making a strong case that normality normalization is robust across conditions. These are all common and accurate choices to backup their claims.

**Other Comments Or Suggestions:**

- A direct side-by-side experiment with at least one of the alternative norms that also claim small-batch stability (e.g., FRN or SwitchableNorm) might strengthen the empirical section.
- Consider using a baselines that are reported in other papers, namely ResNet or ViTs, to make sure that the baselines are not sub-optimal

**Other Strengths And Weaknesses:**

Strengths:

- Straightforward to implement: only adding a power transform step plus scaled Gaussian noise.
- Motivated theoretically very well, the idea of having Gaussian activations is quite an interesting approach and this paper takes an important step in that direction
- Clear discussion of theoretical motivations (information-theoretic and statistical underpinnings).

Weaknesses:

- as mentioned earlier, the paper gives somewhat questionable baselines in Tables 1 & 2. I might be wrong but if
- The experiments in the robust training seem rather thin
- It would be interesting to account for the training overhead from computing  𝜆 in  each iteration in comparison to standard BN

**Questions For Authors:**

- Have authors thought about connection to other domains, namely loss of plasticity in continual learning? Namely, in preserving diversity of hidden units and features during prolonged training

**Relation To Broader Scientific Literature:**

- Normalization layers: The paper gives a lot of focus to batch and layer normalization, which is well deserved, they also mention some other key works such as Decorrelated BN or Switchable Whitenning.  They also cite crucial references on maximum-entropy properties of the Gaussian distribution (Cover & Thomas), Box–Cox and Yeo–Johnson transforms, and typical random smoothing (Cohen et al.).
- Whitening/Orthogonalization: Methods like Decorrelated Batch Normalization (Huang et al.) or Iterative Normalization attempt to whiten features, which is tangentially similar to making them normal. The authors focus on univariate normality; whitened or orthonormal constraints tackle correlations as well.
- Robustness-Enhancing Approaches: The paper cites Randomized smoothing (e.g., Salman et al. 2019), which can provide certified $\ell_2$-robustness via Gaussian noise injection at inference. data augmentation (mixup, AugMix) also yield stable, noise-resistant features. Comparing normality normalization to these could further elucidate how it complements or surpasses them.

**Theoretical Claims:**

- The lemma (Appendix B) that in the bivariate normal case, uncorrelatedness implies independence and also that normality minimizes mutual information for given correlation. This is known in standard information-theoretic references and is stated correctly.
- The second-order series expansion for the negative log-likelihood (Appendices C & D) and the single-step Newton–Raphson for estimating  λ looks coherent to me. authors also provide empirical checks

---

> ### Author Rebuttal · Authors · 2025-03-31
>
> Dear Reviewer 7TPa,
>
> We address all of your comments below.
> >
> >Regarding the baseline performance levels, and the code snippet you provided.
> >
> To address your inquiry regarding the baselines, we ran experiments with the additional use of mixup (Zhang et al. 2017) for several of the model & dataset combinations listed in Table 1 (with the experimental setup otherwise identical to that listed in Appendix E.2) across $M=6$ random seeds. The results are as follows:
> |Dataset|LN|LNN|
> |-|-|-|
> |CIFAR10|89.97 $\pm$ 0.16|**91.18 $\pm$ 0.13**|
> |CIFAR100|66.40 $\pm$ 0.42|**70.12 $\pm$ 0.22**|
> |Food101|73.25 $\pm$ 0.19|**79.11 $\pm$ 0.09**|
>
> These results precisely address your inquiry regarding the baseline performance levels, and provide further substantiating evidence that models trained with normality normalization continue to improve with the use of additional techniques, and consistently outperform other normalization layers.
>
> To further supplement this, please also see our rebuttal comment to Reviewer ps2s regarding the performance of ResNet18 x CIFAR10 with data augmentations, showing again the improvement in performance of BNN.
>
> Next we address the code snippet provided and the performance therein. We actually ran the code you provided, replacing BatchNorm2d with BatchNormalNorm2d: we were able to obtain a performance of $90.95$% which - especially when considering the small number of training epochs ($15$) used in the code snippet - is a significant improvement over the figure you quoted (This is also significant because it is very reasonable to expect that the difference in performance would likely grow with further number of training iterations, because generally techniques using stochastic regularization (such as our additive Gaussian noise with scaling, or as seen elsewhere ex: with the use of dropout) tend to improve more with number of training iterations).
>
> This adds to the evidence that, across a wide array of experimental setups - for example here in the code snippet you provided the optimizer employed (Adam) is different than in our ResNet experiments (SGD), the LR scheduler is different (OneCycleLR vs StepLR), gradient clipping is employed whereas it is not in our setting, and other aspects - that normality normalization consistently outperforms other normalization layers.
>
> Altogether these experiments serve to substantively address your inquiry regarding the baseline performance levels.
> >
> >Regarding a direct side-by-side experiment with an alternative normalization layer.
> >
> We have addressed this as follows: Since we had already invoked decorrelated BatchNorm (DBN) in the text of our paper, we ran experiments using DBN, and compared this with the implementation we developed for decorrelated BatchNormalNorm (DBNN). The experimental setup is consistent with Appendix E.1, and $M=6$ random seeds are also used throughout:
> |Dataset|Model|DBN|DBNN|
> |-|-|-|-|
> |CIFAR10|RN18|90.66 $\pm$ 0.05|**91.50 $\pm$ 0.03**|
> |CIFAR100|RN18|65.11 $\pm$ 0.06|**67.53 $\pm$ 0.10**|
> |STL10|RN34|66.76 $\pm$ 0.29|**69.36 $\pm$ 0.14**|
>
> These results demonstrate a consistent improvement of DBNN over DBN.
> >
> >Regarding accounting for the training overhead from computing $\hat{\lambda}$.
> >
> We believe our Subsection A.5 Speed Benchmarks and Figure 11, referenced in the main body of the text in Subsection 4.6 Additional Experiments & Analysis under paragraph Speed benchmarks, does just this. We have evaluated both the training time and test time overheads.
> >
> >Regarding additional noise robustness experiments.
> >
> Here we provide additional noise robustness results for ResNet18 x CIFAR100:
> |$\phantom{-}$|$\phantom{-}$|L5|L9|L13|L17|
> |-|-|-|-|-|-|
> |**L1**|BNN|**0.047 $\pm$ 0.002**|**0.074 $\pm$ 0.001**|**0.100 $\pm$ 0.002**|**0.386 $\pm$ 0.005**|
> ||BN|0.166 $\pm$ 0.005|0.316 $\pm$ 0.006|0.410 $\pm$ 0.008|1.881 $\pm$ 0.026|
> |**L5**|BNN||**0.027 $\pm$ 0.002**|**0.040 $\pm$ 0.003**|**0.155 $\pm$ 0.012**|
> ||BN||0.069 $\pm$ 0.007|0.088 $\pm$ 0.006|0.438 $\pm$ 0.030|
> |**L9**|BNN|||**0.043 $\pm$ 0.000**|**0.149 $\pm$ 0.002**|
> ||BN|||0.061 $\pm$ 0.001|0.250 $\pm$ 0.003|
> |**L13**|BNN||||**0.258 $\pm$ 0.002**|
> ||BN||||0.396 $\pm$ 0.011|
>
> This provides even further evidence for the findings we presented in Subsection A.3 Noise Robustness.
> >
> >Regarding the suggested references you mentioned.
> >
> We have now included a discussion on the related normalization layers you listed: weight normalization, filter response normalization, normalization propagation, as well as iterative normalization and EvoNorm.
>
> Regarding the copula-based, inverse normal transform, and Lambert W transformation approaches to gaussianizing: we agree these are very interesting avenues for exploration, and have now included a discussion on them in Section 6 Related Work & Future Directions.
>
> We believe we have comprehensively addressed your comments here. We would be highly appreciative if you would consider increasing the score for our submission; thank you.

---

> > ### Comment · Reviewer_7TPa · 2025-04-05
> >
> > I thank the authors for these clarifications.
> > I do find the rebuttal response convincing and I'm supportive of this paper being accepted. So I will increase my score from 3 to 4 (Accept).
> >
> > I have one pending  a few more questions, if authors can clarify or perhaps discuss later in the manuscript, it might be helpful to future readers.
> >
> > - Because the Normality Normalization layers contain essentially two main components, the power transform and the noise injection, a natural question is, what is the effect of each component in isolation, and what is the effect of them combined? Maybe there is already a table or figure I missed? otherwise, some controlled experiment that will show the effect of each and then their combination, or even better, if the magnitude of noise is controlled on a grid and the power transform, then it becomes clear how the two components contribute to the overall effect?
> >
> > - The authors suggest that just one step of Newton type root finding suffices for finding $\lambda$, is there some experiment on what is the additional benefit of more steps (in temrs of Gaussianity metrics) , tha twould be nice
> >
> > - Finally, from my understanding, power transform method is only an approximate way of ensuring Gaussians, which works best if the issue is the distribution being long tailed. Is my impression accurate? If yes, are there more powerful ways of ensuring Gaussianity in a differentiable way?

---

### Official Review · Reviewer_zhYV · 2025-03-13

**Overall Recommendation:** 2

**Summary:**

The paper proposes a normality normalization that enforces Gaussian feature distribution using a power transform and additive Gaussian noise. The motivation for using the normal distribution is to enhance the model's robustness to random perturbations, improving generalization.

## update after rebuttal

The authors have kindly pointed out the parts of their work which I missed or misunderstood.
However, additional clarifications regarding the information-theoretical content of their paper exposed the lack of strong connection between the method and the information-theoretical framework employed. In the manuscript, it is claimed that

> The normal distribution plays a central role in information theory – it is at the same time the best-case signal and worst-case noise distribution,

> the mutual information game suggests gaining robustness to Gaussian noise is optimal because it is the worst case noise distribution

However, Gaussian noise is only the worst-case noise for maximizing $I(X;X+Z)$ if we restrict the second moments of $X$; for other constraints (like restricted support, mean absolute value, etc.), the worst-case noise and best-case distributions **are not Gaussian**.
Thus, robustness to AGN is not a generally desirable outcome: information theory suggests that there may be other cases, in which robustness to, e.g., uniform noise, should yield better results.

Therefore, I insist on additional theoretical analysis (and ablation on noise distribution) being conducted. Otherwise, the work in its current state is not well-supported by the information theory (as constraints imposed and the corresponding "worst-case noise" seem arbitrary). For more information (and additional concerns), please refer to my final reply.

As a result, I decided to keep my score.

**Claims And Evidence:**

- Supported Claims include improved generalization (Tables 1–2 show accuracy gains over traditional BatchNorm/LayerNorm) and Gaussianity in the features (Q-Q plots and $R^2$ metrics validate increased normality).
- The method’s generality claim ("wherever existing normalization layers are used" from lines 42, 411) suggests validation also on non-CV tasks. However, such results are absent from the work.
- Robustness to adversarial as well as random perturbations ("improving model robustness to random perturbations" from lines 42-43) — are not convincingly substantiated. Without adversarial experiments, these broader claims seem unsupported.

**Essential References Not Discussed:**

While the paper cites classical works, it omits several recent advances. For instance, EvoNorm (Liu et al., 2021) is not discussed despite being a contemporary normalization-activation layer method. Additionally, although Iterative Normalization (Huang et al., 2019) is referenced, a more detailed discussion comparing it with the proposed approach would emphasise the contribution.

The work [8a] also uses noise injection and Theorem 5.1 to achieve Gaussian distribution, and, therefore, is closely related to the method proposed in this manuscript.

[8a] Butakov et al. "Efficient Distribution Matching of Representations via Noise-Injected Deep InfoMax". Proc. of ICLR 2025.

**Experimental Designs Or Analyses:**

The experiments are extensive within the computer vision domain, proving performance on multiple datasets and architectures. Despite this, the experimental design omits comparisons with several state‐of‐the‐art (post-BatchNorm) normalization methods and does not explore non-CV applications, even though the method is promoted as generally applicable.

**Methods And Evaluation Criteria:**

- The proposed method combines a power transform with additive Gaussian noise without introducing extra learnable parameters. The approach is implemented as an augmentation to conventional normalization layers.
- While the method is clear and evaluated on several vision architectures and datasets, there is a notable absence of comparisons to more recent or alternative normalization methods (e.g. EvoNorm [3a], Iterative Normalization [Huang et al., 2019]) beyond the classical baselines.
- The Q-Q plots and $R^2$ metrics do not serve as a proper multivariate Gaussianization metric. Perhaps, special statistical tests should be employed, e.g., the Henze-Zirkle test [3b].

[3a] Liu et al. "Evolving Normalization-Activation Layers". arXiv:2004.02967

[3b] Norbert Henze and Bernd Zirkler. "A class of invariant consistent tests for multivariate normality". Communications in Statistics-theory and Methods, 19:3595–3617, 1990

**Other Comments Or Suggestions:**

Perhaps, the "booktabs" table style should also be used for Table 3.

**Other Strengths And Weaknesses:**

The impact of BNN on the runtime compared to BN is not very significant, which is a merit of the method.

**Questions For Authors:**

1. Could the authors rigorously prove theoretical bounds on approximation error or provide another theoretical analysis to strengthen their claim that the quadratic approximation is universally valid under the power transform’s Gaussianization? In particular, how does the approximation error change in deeper layers or with non-Gaussian activations? What are potential limitations of your approach in situations where the activations exhibit strong multimodality or heavy skewness?
1. Could you evaluate normality normalization on non-vision tasks to justify the claim of “wherever existing normalization layers are used”?
1. Could you compare your method with alternative normalization methods (e.g. EvoNorm) beyond the classical baselines?
1. How is the maximization of $I(X;Y)$ from Theorem 5.1 enforced?
1. Have you tried leaving only the noise injection (i.e., not applying the power transform)?

**Relation To Broader Scientific Literature:**

The work is positioned as an extension of BatchNorm (Ioffe & Szegedy, 2015) and LayerNorm (Lei Ba et al., 2016) by explicitly aiming to make Gaussian activations building on classical power transforms (Box & Cox, 1964; Yeo & Johnson, 2000). It also relates to previous work on decorrelation and whitening (Chen & Gopinath, 2000) and on noise-based regularization. However, the discussion would benefit from a more thorough comparison with recent normalization methods and an explicit discussion of how this approach differs fundamentally from methods that already induce some form of Gaussianity (e.g, from the work [8a], see **Essential References Not Discussed**).

**Theoretical Claims:**

- The paper’s derivation of the quadratic approximation for the negative log-likelihood (NLL) to estimate the power transform parameter λ is central. It relies on a series expansion around $\lambda_0 = 1$, and assumes activations are locally Gaussian-like near $\lambda_0$. While empirically valid for the tested cases (Figure 14), it risks failure in deeper layers or complex datasets where activations are multi-modal or heavily skewed. Theoretical guarantees are limited to idealized setup, and the method’s robustness depends on the (unverified) assumption that activations are "close enough" to Gaussian. This constitutes a significant weakness if one wishes to claim universality. Therefore, the authors should test on multi-modal datasets and analyze layer-wise approximation quality.
- There is no clear theoretical evidence that $I(X;X+Z)$ is maximized during the training, which is crucial for the application of Theorem 5.1. Therefore, it is unclear whether noise injection serves any meaningful role in this setup (at least, from the Theorem's 5.1 perspective). Please, compare this to the work [8a] from **Essential References Not Discussed**, where the mutual information is maximized explicitly.

---

> ### Author Rebuttal · Authors · 2025-03-31
>
> Dear Reviewer zhYV,
>
> We address all of your comments below.
> >
> >"The Q-Q plots and $R^2$ metrics do not serve as a proper multivariate Gaussianization metric. Perhaps, special statistical tests should be employed, e.g., the Henze-Zirkle test"
> >
> We very kindly note that we in fact already did precisely this in the paper; using the Henze-Zirkle (HZ) test statistic as well. Please see Section A.7. Joint Normality and Independence Between Features, which is referenced in the main text in Section 4.6 under paragraph Normality normalization induces greater feature independence. This precisely addresses your inquiry.
> >
> >Regarding a comparison to an alternative normalization method.
> >
> Please see the experimental results comparing decorrelated BatchNorm (DBN) (which is also closely related to iterative normalization) with decorrelated BatchNormalNorm (DBNN) in the thread with Reviewer 7TPa, which precisely addresses your inquiry here.
> >
> >"Robustness to adversarial as well as random perturbations ("improving model robustness to random perturbations" from lines 42-43) — are not convincingly substantiated."
> >
> We very kindly point out that we did in fact substantiate the claims we made through the experiments in Section A.3 Noise Robustness, which we also referenced in the main text of the paper in Section 4.6 under paragraph Normality normalization induces robustness to noise at test time. Furthermore, we had only mentioned adversarial robustness as it pertains to deep neural networks in general being susceptible to perturbations; we did not explicitly claim robustness to adversarial perturbations in the paper. However, in Section 6 we provide a line of reasoning which suggests greater adversarial robustness may be attainable, given the connection between robustness to random perturbations and adversarial perturbations.
> >
> >Regarding "justify the claim of “wherever existing normalization layers are used”?".
> >
> We would very kindly like to point out that we indeed investigated normality normalization across several normalization layers, as evidenced by Subsection 4.3 and Figure 1, where we also compared it with InstanceNorm and GroupNorm.
> >
> >Regarding exploration of non-CV tasks.
> >
> In terms of application areas, our intent was to be extremely comprehensive in our experiments in a domain of choice. Because we evaluate across several normalization layers, and in general extensively demonstrate the effectiveness of normality normalization, we believe this is a very strong and reliable indicator for the method's success translating to other (non-CV) domains.
> >
> >Regarding the series expansion of the NLL.
> >
> The power transform we use specifically addresses skewed data distributions - please see (Yeo & Johnson 2000) for a detailed investigation. Furthermore, we have indeed assessed the ability for normality normalization to achieve a high degree of gaussianity on complex datasets, as evidenced by Figures 5 & 6, and our work demonstrates that we can achieve better performance by enforcing a unit's pre-activations to be unimodal-normally distributed.
> >
> >Regarding a discussion on alternative normalization layers.
> >
> We have now cited several works on more recent normalization layers in our paper, including EvoNorm and iterative normalization, as well as weight normalization, filter response normalization, and normalization propagation.
> >
> >"There is no clear theoretical evidence that $I\left(X; X+Z\right)$ is maximized during the training, which is crucial for the application of Theorem 5.1."
> >
> We would with excitement like to clarify this point of inquiry: Our discussion of the mutual information term $I\left(X; X+Z\right)$ uses the following argument as motivation for gaussianizing pre-activations: if the pre-activations are gaussianized, then by Theorem 5.1 $I\left(X; X+Z\right)$ is *necessarily* maximized. This is because a Gaussian distributed variable $X$ maximizes $I\left(X; X+Z\right)$ (as shown by the Theorem) relative to any other distribution for $X$. Thus $I\left(X; X+Z\right)$ is maximized when the gaussianity of $X$ is maximized; even if this occurs implicitly.
> >
> >Regarding comparison to work [8a].
> >
> We found the paper you referenced very interesting and have now cited it in our work. Interestingly, it differs from our work due to the aforementioned point; we discuss the implicit maximization of $I\left(X; X+Z\right)$ and use this idea only as motivation in our work for encoding pre-activations using the Gaussian distribution, whereas to the best of our understanding the work you reference develops a framework for explicitly maximizing this term; making the approaches in the two works interestingly quite distinct and quite complementary. We are eager to explore the possible interplay between our work and this work in follow-up work.
>
> We believe we have comprehensively addressed your comments here. We would be highly appreciative if you would consider increasing the score for our submission; thank you.

---

> > ### Comment · Reviewer_zhYV · 2025-04-09
> >
> > I would like to thank the authors for additional clarifications and pointing out the parts which I missed or misunderstood during the review. Below, I reply to the rebuttal provided.
> >
> > 1. I acknowledge the answer to my first question ("Regarding the series expansion of the NLL."). Although the authors provide additional literature on the question regarding skeweness, I see no comments addressing the multimodality problem. I still insist on a proper theoretical investigation of the solution proposed. If no rigorous theoretical results can be achieved, I kindly ask the authors to emphasise that the second-order method is selected due to empirical success mostly, and no theoretical guaranties are provided.
> > 2. I am now even more confused about the information-theoretical part of the work. If Theorem 5.1 is not used to achieve Gaussian distribution, but to justify Gaussianization, several questions arise:
> >    - There are other nosy channels with different optimal distributions. For example, if $Z$ is small, and we restrict $\mathbb{E} |X| = const$, $I(X;X+Z)$ is maximized for Laplace distribution. If we restrict $\text{supp} \\, X = [0;1]$, $I(X;X+Z)$ is maximized for uniform distribution, etc. For more details, please refer to "maximum entropy distributions". Therefore, choosing Gaussianization over achieving other maximum entropy distributions seems arbitrary.
> >    - The motivation behind the Gaussian noise injection is now more obscure. The authors say:
> >       > the mutual information game suggests gaining robustness to Gaussian noise is optimal because it is the worst case noise distribution
> >
> >       However, for other min-max games for $I(X;X+Z)$, Gaussian noise is no longer the worst case, see the previous point.
> >    - Appendix A.1 and A.4 suggest that adding noise (Gaussian noise in particular) is crucial for the accuracy gains. However, as no MI maximization is performed, there are no rigorous theoretical explanation to this phenomenon. Perhaps, some sort of implicit MI maximization is occuring. In my opinion, the authors should explore this and also provide an ablation study on other min-max MI games to support the hypothesis that MI maximization and performance gains due to selecting the optimal distribution are indeed connected.
> > 3. If the work is focused on empirical results, I still believe that other domains should be explored (e.g., NLP).

---

### Official Review · Reviewer_Eseb · 2025-03-16

**Overall Recommendation:** 4

**Summary:**

The paper presents a novel approach to improving the feature representations in deep neural networks by encouraging normality in activations. The authors introduce Normality Normalization (NormalNorm), a normalization technique based on the power transform to Gaussianize feature distributions and enhance robustness through additive Gaussian noise during training. The paper argues that the normal distribution is optimal for encoding information in neural networks, improving generalization and robustness. Extensive experiments demonstrate the superiority of NormalNorm over traditional normalization techniques like Batch Normalization (BatchNorm), Layer Normalization (LayerNorm), Group Normalization (GroupNorm), and Instance Normalization (InstanceNorm) across multiple model architectures and datasets.

**Claims And Evidence:**

The claims made in the submission are supported by clear and convincing evidence

**Essential References Not Discussed:**

I am not familar with this area and unsure whether there are more essential literature should be cited

**Experimental Designs Or Analyses:**

I check the soundness/validity of any experimental designs or analyses

**Methods And Evaluation Criteria:**

The proposed methods and/or evaluation criteria (e.g., benchmark datasets) make sense for the problem or application

**Other Comments Or Suggestions:**

1. To enhance clarity, I recommend adding an explanation for the derivation of Equation (2). Specifically, it would be helpful to outline the reasoning behind this objective function and why it serves as the appropriate optimization target. Providing an intuitive justification—such as its connection to maximizing the Gaussianity of transformed activations or minimizing divergence from a normal distribution—would strengthen the reader’s understanding of its significance within the broader framework of Normality Normalization.

**Other Strengths And Weaknesses:**

**Strengths**:
1. Theoretical Justification: The authors provide a solid information-theoretic foundation for why Gaussianity in activations is beneficial. They reference the mutual information game framework to argue that normal distributions maximize information transmission and robustness.
2. Methodological Novelty: The proposed Normality Normalization combines the power transform for Gaussianization with additive Gaussian noise with scaling, a distinct approach compared to conventional normalization methods (BatchNorm, LayerNorm, etc.).
3. Comprehensive Empirical Validation: The method is tested on multiple architectures (ResNets, Vision Transformers, WideResNets). Evaluations span diverse datasets, including CIFAR-10, CIFAR-100, SVHN, TinyImageNet, and ImageNet. Experiments consider factors such as network width, depth, and batch size, demonstrating that the method generalizes well.
4: Robustness & Generalization Benefits: The paper shows that Normality Normalization enhances test-time robustness to noise. It improves model generalization, often outperforming conventional normalization techniques. Feature representations exhibit greater independence, an attractive property for reducing redundancy.
5. Strong Quantitative Support: Statistical analysis using Q-Q plots demonstrates that activations become more Gaussianized. The impact of the power transform and Gaussian noise is separately analyzed to isolate their contributions.

**Weaknesses**:
1. Computational Overhead: The power transform involves estimating a transformation parameter ($\lambda$) per feature channel, which increases computational complexity. Although the Newton-Raphson method approximates $\lambda$ efficiently, the added computations may slow down training, as evidenced in the runtime benchmarks.
2. Lack of Analysis on Adversarial Robustness: While Normality Normalization improves robustness to random noise, its effectiveness against adversarial perturbations is not fully examined. Given prior studies linking Gaussian robustness to adversarial defense, further testing in this area would be valuable.

**Questions For Authors:**

1.  **Justification of the Optimization Objective in Equation (2)**: Could you provide a more detailed explanation for why Equation (2) is the appropriate objective function? Specifically: What is the intuition behind minimizing this negative log-likelihood (NLL) in the context of Normality Normalization? Does this directly encourage activations to follow a normal distribution, or is there an implicit assumption about the data distribution?
2. **Adversarial Robustness Claims**: The paper suggests that Normality Normalization improves robustness to random noise, which could imply improved adversarial robustness.  Have you tested Normality Normalization against adversarial perturbations (e.g., FGSM, PGD)? If not, do you expect that the method will improve adversarial robustness, and why?
3. **Potential Issue with Baseline Model Choice**: In Table 1, the reported performance of ViT on ImageNet appears significantly lower than that of the standard ViT model, likely due to the smaller model depth and reduced number of attention heads. The improvement of Layer Normality Normalization (LNN) over Layer Normalization (LN) is observed on this smaller ViT model. Why did the authors choose this particular ViT architecture instead of using the standard ViT model configurations commonly used for ImageNet? Have the authors evaluated whether the performance gap between LNN and LN remains consistent for larger ViT models (e.g., ViT-B/16, ViT-L/16)? If the improvement diminishes on larger models, does this suggest that the benefits of Normality Normalization are more pronounced in smaller-scale networks?

**Relation To Broader Scientific Literature:**

The key contributions of the paper are not related to the broader scientific literature

**Theoretical Claims:**

I did not check the correctness of any proofs for theoretical claims

---

> ### Author Rebuttal · Authors · 2025-03-31
>
> Dear Reviewer Eseb,
>
> We address all of your comments below.
> >
> >"While Normality Normalization improves robustness to random noise, its effectiveness against adversarial perturbations is not fully examined."
> >
> and
> >
> >"Adversarial Robustness Claims: The paper suggests that Normality Normalization improves robustness to random noise, which could imply improved adversarial robustness. Have you tested Normality Normalization against adversarial perturbations (e.g., FGSM, PGD)? If not, do you expect that the method will improve adversarial robustness, and why?"
> >
> We would very kindly like to point out that we invoked adversarial robustness only as it pertains to deep neural networks in general being susceptible to perturbations; we did not explicitly claim robustness to adversarial perturbations in the paper. However, in Section 6 Related Work & Future Directions we provide a line of reasoning which suggests greater adversarial robustness may be attainable, given the connection between robustness to random perturbations and adversarial perturbations. Thus we do expect that on average, greater adversarial robustness should be attainable.
> >
> >Regarding the derivation and justification of Equation 2: the NLL.
> >
> It is great that you inquire about this. We originally decided to defer the derivation for the NLL as it can be found in the literature, for example an outline is provided in (Yeo & Johnson 2000), and a sketch is provided in (Hernandez 1979) for the (related) Box-Cox power transform (https://drive.google.com/file/d/1__hvD4GgwSA3aj2OK9eVnlZg9JOmpSMs/view). We have now included the derivation in the appendix of the paper, and we give the idea here: Begin by taking a random variable $H$ (which can be arbitrarily distributed) and apply the power transform to it (or in practice, a data sample taken from $H$) to obtain $X$. We want $X$ to be as normally distributed as possible, which means we want to maximize the likelihood of $X$ under the Gaussian. This is given by taking the log of the Gaussian PDF for $X$ and maximizing it (this is equivalent to minimizing the negative log-likelihood (NLL)). That is, if you take the (negative) log of the PDF of a Gaussian random variable $X$, then substitute for $x$ the power transform as a function of $h$ (with correct consideration for change of variables), what you will obtain is precisely the NLL we have in our paper.
> >
> >Regarding the choice of ViT architecture.
> >
> We chose to use a somewhat smaller-scale ViT architecture to enable our extensive experiments on several datasets, and to enable high precision in the reporting of our experimental results through the multiple random seeds. In fact, we were able to obtain $M=6$ total seeds for the ImageNet experiments post-submission, for each of LNN and LN. The updated results for ImageNet, across these $M=6$ seeds, are:
> |Dataset|LN|LNN|
> |----------|----------|----------|
> |ImageNet Top1|71.54 $\pm$ 0.16|**75.25 $\pm$ 0.07**|
> |ImageNet Top5|89.40 $\pm$ 0.11|**92.23 $\pm$ 0.04**|
>
> These enable even greater confidence in our experimental results on ImageNet.
>
> To further address your inquiry, we ran experiments with the additional use of mixup (Zhang et al. 2017) for several of the model & dataset combinations listed in Table 1 (with the experimental setup otherwise identical to that listed in Appendix E.2).
> |Dataset|LN|LNN|
> |----------|----------|----------|
> |CIFAR10|89.97 $\pm$ 0.16|**91.18 $\pm$ 0.13**|
> |CIFAR100|66.40 $\pm$ 0.42|**70.12 $\pm$ 0.22**|
> |Food101|73.25 $\pm$ 0.19|**79.11 $\pm$ 0.09**|
>
> These results provide strong evidence that models trained with normality normalization continue to improve with the use of additional techniques for improving generalization performance, and that they continue to outperform models trained with other normalization layers. This also demonstrates that the network size is not an obstacle.
>
> Finally, we found that for both small and large width networks, and for small and large depth networks, normality normalization outperforms competing normalization layers. In Section 4.4 Effectiveness Across Model Configurations, in paragraphs Network Width and Network Depth and through Figures 2 & 3 respectively, we provide experimental evidence demonstrating this. We also found this trend to hold true in experiments with various ViT architectures, and we ultimately used the chosen architecture in the paper to enable extensive experiments and with multiple random seeds, as mentioned.
>
> We have furthermore added experimental results contrasting decorrelated batch normalization (DBN) with decorrelated batch normality normalization (DBNN); please see the thread with Reviewer 7TPa. These experimental results provide further evidence for the strong performance of normality normalization across various normalization layers.
>
> We believe we have comprehensively addressed your comments here. We would be highly appreciative if you would consider increasing the score for our submission; thank you.

---

> > ### Comment · Reviewer_Eseb · 2025-04-05
> >
> > Thank you for all the responses. I will revise my score.

---

### Decision · Program_Chairs · 2025-05-01

**Decision:**

Accept (poster)

**Comment:**

The main contribution of this work is demonstrating that Gaussianizing data representations across the layers of deep neural networks can enhance the performance of deep neural networks. This observation provides valuable insights into the inner workings of normalization layers, which are known to induce Gaussian-like distributions at initialization. Interestingly, explicitly enforcing this Gaussian property further improves the performance of convolutional neural networks (CNNs). While there are concerns regarding the depth of insights and theoretical arguments, the experimental contribution is substantial enough for ICML acceptance.